# Decoding Projections From Frozen Random Weights in Autoencoders: What Information Do They Encode?

**Nancy Thomas** *
AI Research, J.P. Morgan Chase & Co.
noniekt@gmail.com

**Keshav Ramani**
AI Research, J.P. Morgan Chase & Co.
keshav.ramani@jpmchase.com

**Annita Vapsi**
AI Research, J.P. Morgan Chase & Co.
annita.vapsi@jpmchase.com

**Daniel Borrajo**
AI Research, J.P. Morgan Chase & Co.
daniel.borrajo@jpmchase.com

## Abstract

Despite the widespread use of gradient-based training, neural networks without gradient updates remain largely unexplored. To examine these networks, this paper utilizes an image autoencoder to decode embeddings from an encoder with fixed random weights. Our experiments span three datasets, six latent dimensions, and 28 initialization configurations. Through these experiments we demonstrate the capability of random weights to capture broad structural themes from the input and we make a case for their adoption as baseline models.

## 1 Introduction

Deep neural networks learn task-specific input representations by defining a loss function and updating weights through gradient calculations. While research has focused on optimizing representations and architectures, the inherent representations that could be formed by a model's architecture without gradient updates have received less attention. As models become larger and more resource-intensive, exploring this aspect is increasingly important. This investigation could potentially reduce the need for gradient updates, improve neural network explainability, and support more sustainable AI practices.

In a fixed weight setting, network weights remain at their initial random values. For a single-layer network: $Y = f(W^T X) + b$, freezing $W$ with random weights focuses on random transformations of $X$ and the activation function $f$. To examine the information preservation of random weights we employ an autoencoder architecture, applying multiple transformations in the encoder with the ReLU activation function. We selected autoencoders for their simple architecture and the perceptible task of reconstruction, which allows visual inspection of results to assess reconstruction success. This approach aligns with recent studies on frozen random weights in CNNs with ReLU activation functions [Nachum et al., 2022a].

Our work raises two key questions:

- How useful are representations generated by random weights in neural networks?
- What information is captured in these representations?

Previous work on random weights has primarily been theoretical. Our study is the first empirical investigation to document and quantify the perceptibility of representations generated with random

---

*All contributions by this author were made while working at AI Research, J.P. Morgan Chase & Co.

weights. It provides empirical evidence of the predictive capacity of random weight networks and highlights the distinction between architecture and training.

## 2   Related Work

The effectiveness of random weights can be understood through various theoretical lenses. The Johnson-Lindenstrauss lemma provides a foundational explanation, showing that random projections approximately preserve pairwise distances with high probability [Freksen, 2021, Nachum et al., 2022a]. For linear fully-connected networks, this geometric preservation is straightforward, while networks with ReLU activations exhibit more nuanced behavior through angle contraction [Nachum et al., 2022a]. Some studies have explored the validity of the Johnson-Lindenstrauss lemma for random matrices with elements drawn from distributions such as Gaussian, Orthogonal, and Hadamard [Johnson and Lindenstrauss, 1984, Ailon and Chazelle, 2009, Achlioptas, 2003]. Further discussion on related works, including the scope and applications of random weights in neural networks, their theoretical foundations, the effect of initializations and training dynamics on performance, and their relationship to kernel methods, is detailed in Section A.1 of the Appendix.

## 3   Method

We limit our study to autoencoders with randomly initialized encoder weights using specific methods/distributions, while decoder weight initializations follow PyTorch's default settings. The list of all initialization methods considered for the encoder are discussed in Table 1 of the Appendix. By default, ConvTranspose2d decoder layers are initialized using Kaiming Uniform initialization [He et al., 2015], where the scale is determined by the number of output channels, while linear layers use the same initialization based on the number of input channels. Biases are initialized to zero for convolutional layers and to a Uniform distribution proportional to the inverse square root of the number of input channels for linear layers. [2] In our implementation, the encoder consists of convolutional layers, followed by the ReLU activation. Our choice is motivated by the works of Nachum et al. [2022b] that discusses the theoretical properties of random weights in convolutional layers with ReLU. The decoder consists of two fully connected layers followed by a set of convolutional layers. Further details can be found in Section A.2 of the Appendix.

Formally, let $X$ denote the input image, $Z$ the latent representation, and $\hat{X}$ the reconstructed image. The encoder is represented by the function $F_{\theta_e}(X) = Z$, where $\theta_e$ are the encoder weights. The decoder is represented by the function $G_{\theta_d}(Z) = \hat{X}$, where $\theta_d$ are the decoder weights.

We consider two experimental scenarios. In the first scenario, both the encoder and decoder weights are updated during training. We refer to this scenario as **Learnable** when analyzing our results. In the second scenario, the encoder weights $\theta_e$ are initialized and frozen, while the decoder weights $\theta_d$ are updated. We refer to this scenario as **Fixed**. The objective for both scenarios is to minimize the reconstruction loss $L(X, \hat{X})$, where $\hat{X} = G_{\theta_d}(F_{\theta_e}(X))$. The reconstruction loss is typically measured using the mean squared error and the weights are updated using gradient descent.

Reconstruction loss measures image similarity but not structural or semantic similarity. Images with low reconstruction errors can still appear dissimilar as demonstrated by the work of Zhang et al. [2018]. To assess structural and perceptual similarity, we use the Structural Similarity Index (SSIM) [Wang et al., 2004] and Frechet Information Distance (FID) [Heusel et al., 2017]. These perceptual loss functions, along with reconstruction loss, evaluate similarity between the generated image $\hat{X}$ and input image $X$, and are used in our experiments (see Section A.4 of the Appendix).

To evaluate the generality of our findings, we varied three parameters: latent space dimensionality, weight initialization method, and encoder weight updates. We tested 8 initialization methods—Gaussian, Orthogonal [Saxe et al., 2013], Uniform, Xavier Uniform, Xavier Normal [Glorot and Bengio, 2010], Kaiming Uniform, Kaiming Normal [He et al., 2015], and Hadamard [Zhao et al., 2022]—resulting in 28 configurations. Six latent space dimensions (16, 32, 64, 128, 256, 512) were tested. For each combination, 5 models were trained with different seeds to mitigate seed-specific effects. We used three datasets (CIFAR-10, CIFAR-100, and Fashion-MNIST) and

---

[2]https://pytorch.org/docs/stable/nn.init.html

experiments involved 840 model pairs per dataset—one with learnable encoder weights and one with fixed weights—totaling 2520 pairs. More information on the three datasets used can be found in Section A.5 of the Appendix. Models were trained on a single GPU from a g4dn.12xlarge instance, with up to 500 epochs and early stopping.

# 4   Results and analysis

In an effort to understand the relationship between fixed and learnable weights, we plot their metrics across all random seeds, latent dimensions, and initializations and calculate the $R^2$ relative to the best-fit line. Through this strategy we discovered at least one parametrization per initialization strategy with a high $R^2$ and slope around 0.8. We select these parametrizations where fixed weight metrics scale linearly with learnable weight metrics. Table 2 in the Appendix Section A.6 presents the average $R^2$ and slope of the best-fit line for each configuration.

Figure 1 displays plots for the selected parametrizations with the best-fit line. A broad linear relationship is observed between learnable and fixed networks as latent dimension increases, evaluated using reconstruction loss, SSIM, and FID. This trend persists until dimensions reach 128, after which occasional outliers appear. Reconstruction loss and FID show a general correlation across datasets, while SSIM exhibits a weaker correlation, potentially due to SSIM emphasizing exact structural similarity that is easily disrupted in reconstructions.

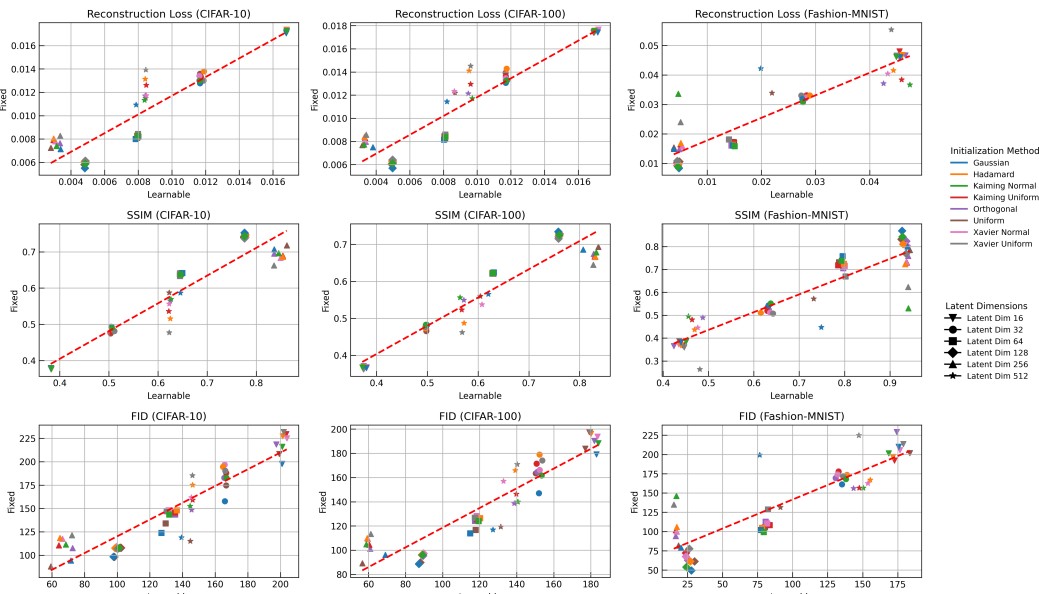

Figure 1: **Fixed versus learnable scenarios for different metrics** The color family represents the distribution used and the shape stands for the latent dimension. The red dashed line in every plot represents the line of best fit.

Figure 2 shows average performance values across random seeds against latent dimensions for selected parametrizations. In CIFAR-10, CIFAR-100, and Fashion-MNIST, learnable encoder weights lead to decreasing reconstruction loss as latent dimension increases until around 256, then gradually rises. Fixed weights show similar behavior, with loss decreasing until around dimension 128. Learnable weights generally outperform fixed weights, but differences are minimal below dimension 128 and converge above 256. This is expected, as learnable encoders can better represent images for the decoder, while fixed encoders rely on the decoder to interpret random projections. Fixed weights follow the trajectory of learnable weights across dimensions, showing reasonable performance.

For SSIM, where higher values are better, a similar trend is observed with learnable weights outperforming fixed weights, especially at higher latent dimensions. For FID, which assesses feature space similarity, the trend is similar, but the difference between learnable and fixed settings is smaller

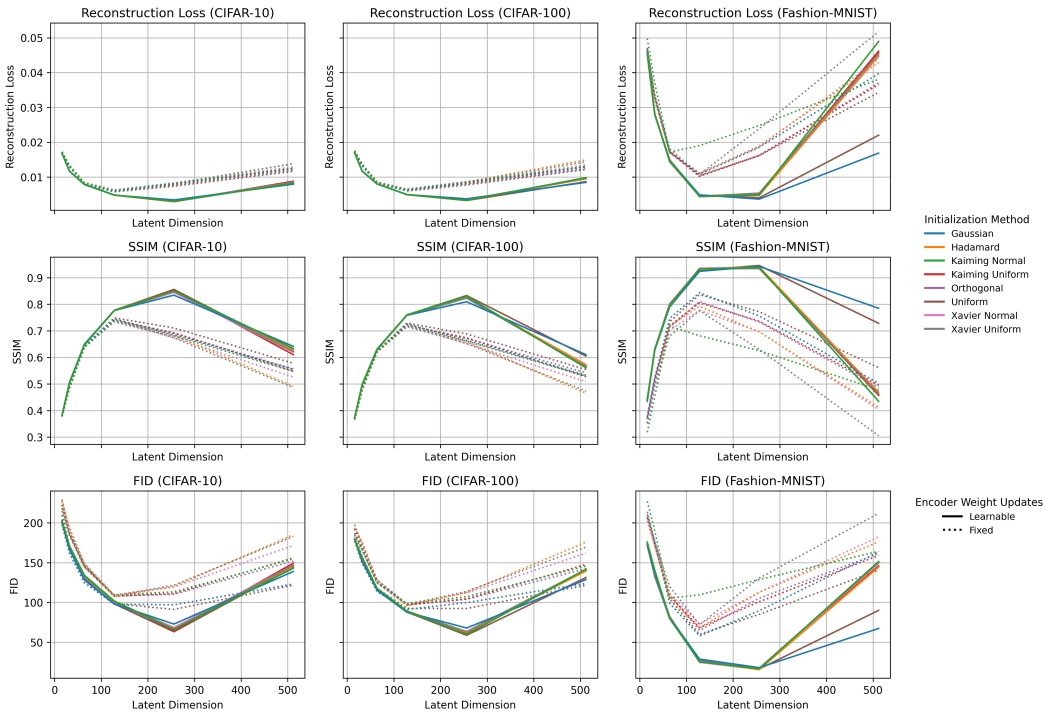

Figure 2: **Average metrics across latent dimensions** Average reconstruction error, SSIM, and FID across 5 random seeds for each weight initialization method for both learnable and fixed weights.

at higher latent dimensions. This may be because FID is more sensitive to image blurriness than reconstruction loss (see Appendix Section A.7), where images appear blurry in both settings.

Fashion-MNIST exhibits similar trends to CIFAR-10 and CIFAR-100, with minor differences. Due to its single-channel nature, the reconstruction loss reaches a minimum more quickly than in three-channel datasets. At high latent dimensions, learnable models sometimes have slightly worse reconstruction loss than fixed models, though both perform poorly. SSIM and FID plots show similar behavior, with FID consistently better in the learnable setting compared to the fixed setting. A qualitative analysis (see Appendix A.7) also confirms the finding that fixed weights are able to capture meaningful representations comparable to leanable weights. Results for all dataset, metric and parameter configuration combinations are available in Section A.8 of the Appendix.

Overall, training models with fixed weights is faster and less computationally intensive than using fully learnable weights. For instance, training a model with a latent dimension of 512 on CIFAR-10, CIFAR-100, and Fashion-MNIST took 7657, 3874, and 8616 seconds, respectively, in the learnable setting, compared to 5177, 2047, and 3066 seconds in the fixed setting.

## 5    Discussion and Conclusion

Our experiments have led to two significant insights that deepen our understanding of neural networks. Firstly, we empirically demonstrate that fixed random weights are capable of capturing structural themes within inputs even without gradient updates. This suggests that these weights can serve as effective baselines for exploring the representational capabilities of neural networks. Secondly, the use of fixed random weights provides a foundation for efficient architecture selection, as there is a linear relationship between losses in settings with fixed weights and those with learnable weights. These findings highlight the baseline representational ability of neural networks and underscore the potential of utilizing frozen random weight initializations to enhance their performance and efficiency. Our work complements and confirms the growing body of work similar to Nachum et al. [2022b] as perceptible empirical demonstrations.

A limitation of our work is testing fixed random weights on only three datasets: CIFAR-10, CIFAR-100, and Fashion-MNIST, which may not capture real-world image variation. While latent dimensions varied, network width and depth were unchanged. Training was capped at 500 epochs, with most models converging early. Future work could explore different activations, generalize findings to other architectures and initialization functions, and deepen the theoretical understanding of random weights in neural networks.

# 6   Disclaimer

This paper was prepared for informational purposes by the Artificial Intelligence Research group of JPMorgan Chase & Co. and its affiliates ("JP Morgan") and is not a product of the Research Department of JP Morgan. JP Morgan makes no representation and warranty whatsoever and disclaims all liability, for the completeness, accuracy or reliability of the information contained herein. This document is not intended as investment research or investment advice, or a recommendation, offer or solicitation for the purchase or sale of any security, financial instrument, financial product or service, or to be used in any way for evaluating the merits of participating in any transaction, and shall not constitute a solicitation under any jurisdiction or to any person, if such solicitation under such jurisdiction or to such person would be unlawful.

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

# A    Appendix

In this appendix, we provide a review on related works and a quantitative and qualitative results of various configurations of initializations discussed earlier. By doing so, we aim to provide a comprehensive understanding of the effects of fixed random weights in generating representations of input images.

## A.1    Related Work

In some of the earliest works, randomly sampled weights were found to be suitable for simple tasks when used with perceptrons [Hebb, 1949, Minsky and Selfridge, 1961, Minsky, 1963]. With the evolution of perceptrons to neural networks, random weights have continued to demonstrate surprising effectiveness across various tasks and architectures. For instance, random weights have been used to show that the last layer of a neural network is more important than the hidden layers [Schmidt et al., 1992]. The effectiveness of random weights was studied using the Random Vector Functional Link networks [Pao et al., 1994], three-layer (input, hidden, output) feedforward networks with fixed random weights in the hidden layer and direct input-output connections. Extreme Learning Machines (ELMs), a type of neural network in which the hidden layer weights are randomly initialized and never updated, were shown to be theoretically and empirically capable of learning concepts [Huang et al., 2004]. SVMs [Vapnik, 1995] and Least Square SVMs [Suykens and Vandewalle, 1999] were found to perform well with ELMs as well [Huang et al., 2011].

Beyond the initial work on random weight networks, the success of random weights has been observed across more modern network architectures including convolutional neural networks [Jarrett et al., 2009, Saxe et al., 2011], attention mechanisms [Fu et al., 2023], and traditional feedforward networks [Giryes et al., 2016]. Further studies expanded this understanding, demonstrating success with partially frozen weights [Rosenfeld and Tsotsos, 2019], networks with only learnable biases [Williams et al., 2024], and even single-hidden-layer feed forward networks (SLFNs) with random hidden nodes [Huang et al., 2006]. Our approach deviates from existing literature by utilizing a randomly initialized and partially frozen autoencoder network. This is done with the aim of enhancing the understanding of the performance of random neural networks. By reconstructing the input images, the autoencoder allows us to visually assess how much information is preserved, making the network's performance perceptible to the eye.

**Scope and Applications** The breadth of applications spans from object recognition [Jarrett et al., 2009] to representation inversion [He et al., 2016, Ulyanov et al., 2018] and texture synthesis [He et al., 2016]. In computer vision specifically, Jarrett et al. [2009] achieved a 63% recognition rate on Caltech-101 using random filters, while Rosenfeld and Tsotsos [2019] showed that learning only a small subset of network parameters or layers leads to surprisingly minimal performance degradation. They suggest that this may be a result of significant over-parametrization in current models. The versatility extends to bias-learned networks in auto-regressive modeling, multi-task learning, and dynamical system forecasting [Williams et al., 2024], highlighting the broad applicability of random weight approaches in neural networks. In the language domain, random weights have been used for effective sentence embeddings [Wieting and Kiela, 2019] and abstractive summarization [Pilault et al., 2020]. Our work focuses on a vision reconstruction task using a partially frozen autoencoder. We believe that the granularity of reconstruction metrics and the ability to visually demonstrate the extent of information preservation in random initializations provide empirically useful results, which are not extensively covered in the existing literature.

We find that some works, such as He et al. [2016], have explored generating meaningful visual representations of input images by inverting feature representations using generative techniques. These approaches reconstruct images through iterative refinement, starting from white noise inputs, rather than relying on an autoencoder with random and frozen encoder weights. Similarly, in the context of the work of Ulyanov et al. [2018], the goal is to recover the original image from a degraded or incomplete version by using a randomly-initialized generator network as a handcrafted prior. In contrast, our approach aims to capture the amount of information preserved by a random weight encoder both qualitatively and quantitatively.

**Theoretical Foundations** The effectiveness of random weights can be understood through multiple theoretical lenses. The Johnson-Lindenstrauss lemma provides a foundational explanation, demonstrating that random projections approximately preserve pairwise distances with high proba-

bility [Freksen, 2021, Nachum et al., 2022a]. For linear fully-connected networks, this geometric preservation is direct, while networks with ReLU activations exhibit more nuanced behavior through angle contraction [Nachum et al., 2022a]. Some works in the literature explored the validity of the Johnson-Lindenstrauss lemma for random matrices with elements drawn from various distributions including Gaussian, Orthogonal and Hadamard distributions Johnson and Lindenstrauss [1984], Ailon and Chazelle [2009], Achlioptas [2003].

Building on these geometric insights, Giryes et al. [2016] proved that deep neural networks with random i.i.d Gaussian weights produce distance-preserving embeddings, with particular emphasis on in-class versus out-of-class data discrimination. The recent theoretical contributions of Williams et al. [2024] have further strengthened these foundations, proving that networks with fixed random weights but learnable biases can approximate arbitrary functions with high probability. This extends to SLFNs, where Huang et al. [2006] demonstrated that input weights and hidden layer biases need not be tuned at all for universal approximation, provided appropriate activation functions are chosen. We regard these works as fundamental in explaining why random weights perform well. We extend this foundational research by empirically and perceptually validating their findings using various weight initialization distributions and non-linear neural network encoder-decoder architectures for autoencoders.

**Initialization and Training Dynamics** Research has shown that initialization significantly impacts training dynamics [Hanin and Rolnick, 2018], with poor initializations leading to more frequent training failures in deeper networks. The mean and variance of length scales strongly predict early training dynamics, and proper initialization of weights and residual modules prevents exponential growth or decay of activation sizes with depth. Furthermore, Saxe et al. [2013] introduced the concept of dynamical isometry for faithful backpropagation, demonstrating that, while random Gaussian initializations cannot achieve this condition, random orthogonal initializations can, enabling depth-independent learning times. Finally, in the intersection of random weights with neural architecture search [Gaier and Ha, 2019], the discovery of "lottery tickets" [Frankle and Carbin, 2018] and "supermasks" [Zhou et al., 2019] suggests that the initialization plays a crucial role in network performance, opening new avenues for understanding the relationship between network architecture, initialization, and training. Our work supports the importance of initialization for feature preservation over random weights and Section 4 addresses this phenomenon by examining the performance of the partially randomly initialized autoencoder using an array of initialization distributions.

**Relationship to Kernel Methods** The connection between random weights and kernel methods offers another theoretical perspective. Random feature approaches, originally proposed as computationally efficient alternatives to kernel methods [Rahimi and Recht, 2007], have shown that randomized feature maps can effectively approximate kernel functions. Rahimi and Recht [2008] demonstrated comparable performance between shallow architectures with random non-linearities and those with optimally tuned ones.

**Limitations and Considerations** Despite these successes, important limitations exist. Yehudai and Shamir [2019] proved that random features cannot efficiently learn even a single ReLU neuron over standard Gaussian inputs unless the network size or weight magnitude is exponentially large. Additionally, Li et al. [2023] have derived new approximation error lower bounds for depth-2 band-limited random neural networks, showing that, when hidden parameters are distributed in a bounded domain, zero approximation error may not be achievable.

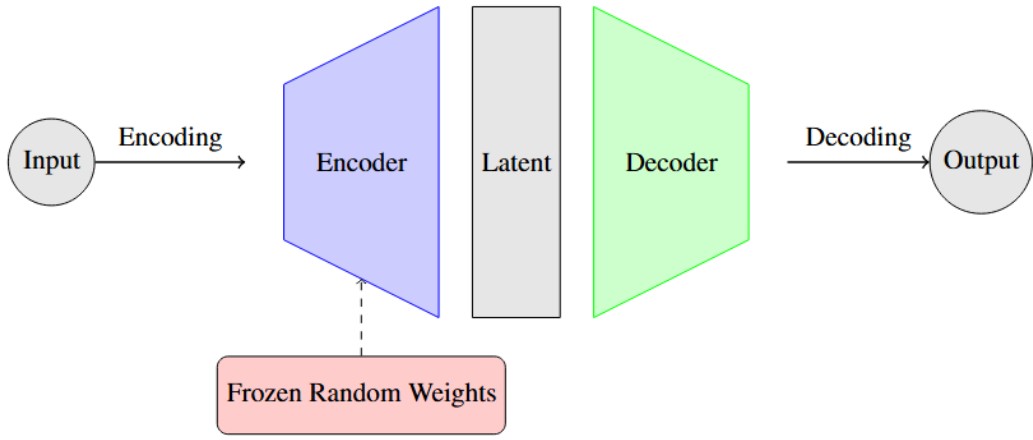

Figure 3: An overview of the experimental setup and architecture.

## A.2  Architecture

This section discusses the design of the encoder and decoder used in our experiments. Most of our design choices (as previously highlighted) are motivated by prior works.

### A.2.1  Encoder

In our experiments, the encoder exclusively consists of convolutional layers followed by the ReLU activation function. Theoretical works of Nachum et al. [2022b] have been centered around such layers and therefore this choice was made. The convolutional layers progressively reduce the dimensions of the input image and therefore learn a compact and dense representation of the input.

### A.2.2  Decoder

The decoder in each experiment consists of two fully connected learnable linear layers followed by learnable convolutional layers with the ReLU activation function. This choice was made with the idea of the decoder learning encoder specific representations using the linear transformations followed by convolutional operations.

While we explore this controlled setting, we acknowledge that future works could expand on the architectures in scope.

## A.3  Neural network weight initialization methods

Table 1: Neural network weight initialization methods. We experiment with the below initialization methods. These were chosen mainly for their popularity and ubiquity. For each initialization, we experiment with several parametrizations. The complete results can be found in Section A.8 of the Appendix.

| Initialization Method | Equation/Description | Tested Configurations | Notes |
|---|---|---|---|
| Gaussian (Normal) | $W \sim \mathcal{N}(\mu, \sigma^2)$ | $(\mu, \sigma)$: (0, 1), (0, 0.5), (0, 0.02), (1, 1), (1, 0.5), (1, 0.02), (-1, 1), (-1, 0.5), (-1, 0.02) | Weights are drawn from a Normal distribution with mean $\mu$ and variance $\sigma^2$. |
| Uniform | $W \sim \mathcal{U}(a, b)$ | $(a, b)$: (-1, 1), (-0.5, 0.5), (-0.02, 0.02), (0, 1), (-1.5, 1.5) | Weights are drawn from a Uniform distribution between $a$ and $b$. |
| Xavier (Glorot) Uniform | $W \sim \mathcal{U}\left(-\frac{\sqrt{6}}{\sqrt{n_{in}+n_{out}}}, \frac{\sqrt{6}}{\sqrt{n_{in}+n_{out}}}\right)$ | Gain: 1, 0.5, 1.5 | Suitable for sigmoid and tanh activations. Gain scales the standard deviation of the weight initialization. |
| Xavier (Glorot) Normal | $W \sim \mathcal{N}\left(0, \frac{2}{n_{in}+n_{out}}\right)$ | Gain: 1, 0.5, 1.5 | Suitable for sigmoid and tanh activations. Gain scales the standard deviation of the weight initialization. |
| Kaiming (He) Normal | $W \sim \mathcal{N}\left(0, \frac{2}{n_{in}}\right)$ | Mode: fan_in, fan_out | Suitable for ReLU and variants. Mode controls weight normalization using input or output neurons. |
| Kaiming (He) Uniform | $W \sim \mathcal{U}\left(-\sqrt{\frac{6}{n_{in}}}, \sqrt{\frac{6}{n_{in}}}\right)$ | Mode: fan_in, fan_out | Suitable for ReLU and variants. Mode controls weight normalization using input or output neurons. |
| Orthogonal | $W = Q$ | Gain: 1, 0.5, 1.5 | $Q$ is an orthogonal matrix obtained from the QR decomposition of a random matrix. Gain scales the standard deviation of the weight initialization. |
| Hadamard | $W = H$ | Default configuration only | $H$ is a Hadamard matrix with random sign flips and scaling to control weight variance. |

## A.4 Metrics Definitions

### A.4.1 Structural Similarity Index (SSIM)

SSIM is defined as:

$$\text{SSIM}(X, \hat{X}) = \frac{(2\mu_X \mu_{\hat{X}} + C_1)(2\sigma_{X\hat{X}} + C_2)}{(\mu_X^2 + \mu_{\hat{X}}^2 + C_1)(\sigma_X^2 + \sigma_{\hat{X}}^2 + C_2)}$$

where $\mu$ and $\sigma$ represent the means and standard deviations, and $C_1$ and $C_2$ are constants to stabilize the division.

### A.4.2 Frèchet Inception distance (FID)

FID is defined as:

$$\text{FID}(X, \hat{X}) = \|\mu_X - \mu_{\hat{X}}\|^2 + \text{Tr}(\Sigma_X + \Sigma_{\hat{X}} - 2(\Sigma_X \Sigma_{\hat{X}})^{1/2})$$

where $\mu$ and $\Sigma$ are the means and covariances of the feature vectors extracted from the images using a pretrained Inception-v3 model and Tr indicates the trace operation.

## A.5  Data

We used three datasets: CIFAR-10, CIFAR-100 [Krizhevsky, 2009], and Fashion-MNIST [Xiao et al., 2017]. CIFAR-10 has 60,000 color images (32x32 pixels) across 10 classes, with a default split of 50,000 training and 10,000 test images. We allocated 20% of the training images for validation. CIFAR-100 also contains 60,000 color images (32x32 pixels) but across 100 classes, following the same train-test split and validation allocation. Fashion-MNIST includes 70,000 grayscale images (28x28 pixels) in 10 classes, with a similar split and validation setup. These datasets are widely used in computer vision, making them suitable for our experiments. [3].

## A.6  Analysis of Fixed Versus Learnable Metrics

In Figure 1, we plot the fixed versus learnable value for each of our three metrics and for each of our three datasets. As shown in this figure, many weight initialization configurations seem to display a linear relationship between the fixed and learnable metrics. For each initialization configuration, we examine the line of best fit through these points, as well as the $R^2$ to the line of best fit. The average of these value across all metrics and datasets, along with the average slope of the associated line of best fit, are recorded in Table 2. Within each initialization family, the configuration with the highest $R^2$ is shown in bold. These values tend to be high, indicating that for each initialization, there exists a configuration for which the relationship between fixed and learnable losses is approximately linear.

---

[3]ImageNet was excluded due to licensing restrictions, leaving it for future work

Table 2: **Learnable versus fixed metrics** The average $R^2$ and average slope of the lines of best fit through the fixed versus learnable points for all weight initialization configurations. The average is taken across all three metrics and all three datasets and the configuration with the highest $R^2$ within each initialization family is shown in bold.

| Initialization Method | Average $R^2$ | Average slope |
|---|---|---|
| Gaussian ($\mu = -1, \sigma = 0.02$) | 0.6878 | 0.801 |
| Gaussian ($\mu = -1, \sigma = 0.5$) | 0.0997 | 0.0272 |
| Gaussian ($\mu = -1, \sigma = 1$) | 0.0389 | -0.0211 |
| Gaussian ($\mu = 0, \sigma = 0.02$) | 0.6018 | 0.6704 |
| Gaussian ($\mu = 0, \sigma = 0.5$) | **0.8315** | 0.7963 |
| Gaussian ($\mu = 0, \sigma = 1$) | 0.3186 | 2.1667 |
| Gaussian ($\mu = 1, \sigma = 0.02$) | 0.5559 | 1.0433 |
| Gaussian ($\mu = 1, \sigma = 0.5$) | 0.0863 | -0.6091 |
| Gaussian ($\mu = 1, \sigma = 1$) | 0.3174 | -2.6777 |
| Orthogonal (Gain=0.5) | 0.2123 | 0.7083 |
| Orthogonal (Gain=1) | 0.8731 | 0.7531 |
| Orthogonal (Gain=1.5) | **0.898** | 0.7583 |
| Uniform (-1.5, 1.5) | 0.3812 | 2.2416 |
| Uniform (-1, 1) | 0.8474 | 0.8438 |
| Uniform (-0.5, 0.5) | **0.9036** | 0.8335 |
| Uniform (-0.02, 0.02) | 0.1462 | 0.2744 |
| Uniform (0,1) | 0.092 | 0.5189 |
| Xavier Normal (Gain=0.5) | 0.0673 | -0.4711 |
| Xavier Normal (Gain=1) | 0.4361 | 0.7774 |
| Xavier Normal (Gain=1.5) | **0.8827** | 0.7826 |
| Xavier Uniform (Gain=0.5) | 0.1524 | -1.0842 |
| Xavier Uniform (Gain=1) | 0.2027 | 0.6493 |
| Xavier Uniform (Gain=1.5) | **0.8388** | 0.7935 |
| Kaiming Normal (Mode=fan_in) | **0.7971** | 0.7012 |
| Kaiming Normal (Mode-fan_out) | 0.5151 | 0.5359 |
| Kaiming Uniform (Mode=fan_in) | **0.8935** | 0.7766 |
| Kaiming Uniform (Mode=fan_out) | 0.5534 | 0.7364 |
| Hadamard | **0.843** | 0.7894 |

## A.7 Qualitative comparison of image reconstructions

While several works have discussed the theoretical aspects of working with random weights in neural networks, seldom have they explored the perceptibility of results generated using them. In this section we explore this aspect of the images reconstructed in the fixed scenario and compare the results arising from different initializations.

Figures 4, 5, and 6 show samples of the reconstructed images produced by the fixed and learnable scenarios from various initializations for the CIFAR-10, CIFAR-100, and Fashion-MNIST datasets respectively. We show the reconstructions for the initializations that have demonstrated the highest $R^2$ for each distribution as outlined in Table 2.

We notice that across datasets, the sampled reconstructions from the fixed scenario are all perceptibly comparable to the ones from the learnable scenario. In Figure 4 we notice broad themes of the inputs being captured by the fixed random weights and the shapes of the cars, objects, and animals are well outlined. The low resolution of images from the CIFAR-10 dataset hinders a closer commentary on other features of the input. In CIFAR-10 as well as CIFAR-100, we notice colors and shapes of multiple artifacts in the image being reconstructed from the embeddings generated in the fixed scenario. This trend carries over to the monochromatic images shown in Figure 6 where the reconstructions are comparable between both scenarios. Features of the input image such as textures, patterns, and

outline of texts are also shown to be captured in the representations generated by the fixed weight encoder.

In summary, these figures sufficiently demonstrate that some random weight initializations, when fixed, are able to generate reasonable embeddings from which the input image can be decoded. It must also be noted that a few fixed and learnable initializations tend to have poor quality reconstructions. However, the better performing parameters of each distribution seem to be captured well by the highest $R^2$ values from Table 2, as can be qualitatively confirmed from these figures.

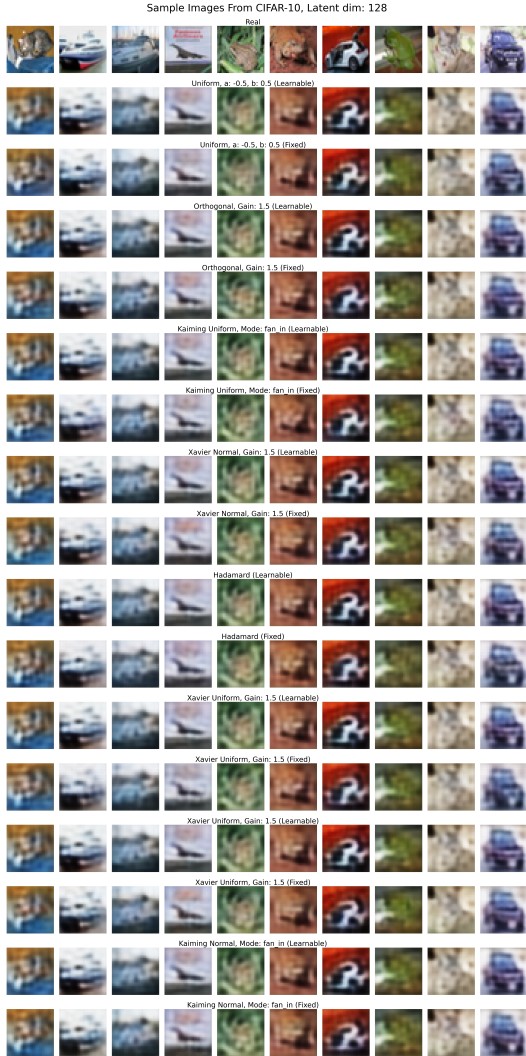

Figure 4: **CIFAR-10 reconstructions** Visualization of the reconstruction of images from the latent dimension 128 for the CIFAR-10 dataset for various initializations in both the fixed and learnable setting.

## A.8 Analyzing different parameters for random distributions

In this section we document the results from reconstruction loss, SSIM, and FID for different configurations of the distributions in consideration. In order to be comprehensive, we study multiple parameters for each distribution, and their results can be found below. All results are the average of five random seeds as noted earlier and this section is not limited to the models with the best $R^2$ scores alone. For each latent dimension, the best value across initializations for either fixed or learnable is shown in bold.

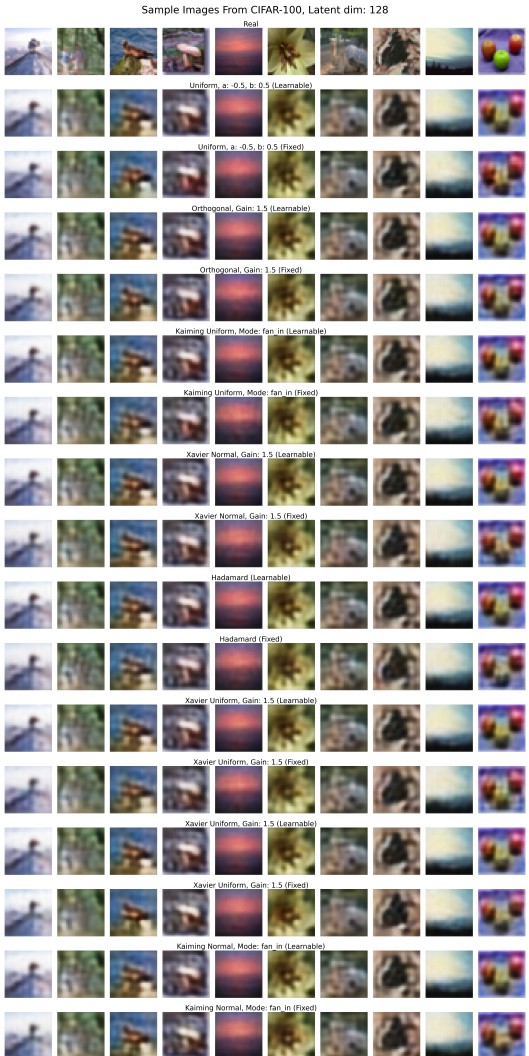

Figure 5: **CIFAR-100 reconstructions** Visualization of the reconstruction of images from the latent dimension 128 for the CIFAR-100 dataset for various initializations in both the fixed and learnable setting.

### A.8.1   CIFAR-10

Tables 3, 4, and 5 tabulate the reconstruction loss, SSIM, and FID for the CIFAR-10 dataset respectively. From Table 3, we notice that learnable weights consistently achieve a lower reconstruction loss, with no single initialization demonstrating the best performance. This is not a surprise as the encoders and decoders in the learnable scenario were trained to yield the lowest reconstruction loss.

This trend seems to have mostly carried over to the evaluations using SSIM, as evidenced by Table 4 with the only exception being $dim = 16$. By a narrow margin, we note that the fixed scenario is able to produce images with the highest structural similarity with the input in this setting alone.

However, with the FID we notice the fixed scenario performs consistently better than the learnable scenario. This is noteworthy, as the embeddings generated in the fixed scenario had no information about the reconstruction task. The decoder was the only learning component, and seem to have reconstructed images with high similarity to the input as measured by the FID. The only exception to this trend is the setting where $dim = 256$ in which the learnable scenario achieved the least FID. Further we notice that the uniform initialization generally performs better than other initializations, followed by the Gaussian initialization.

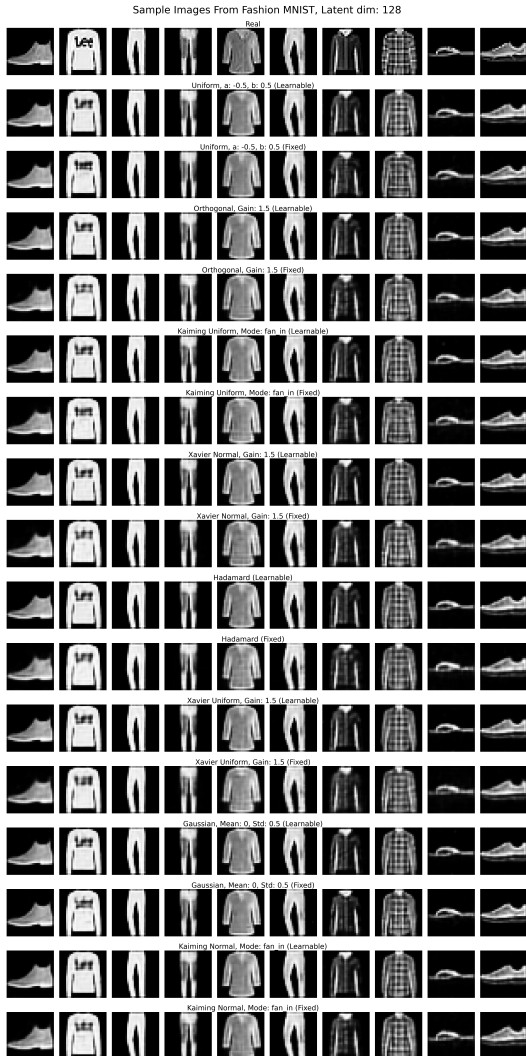

Figure 6: **Fashion-MNIST reconstructions** Visualization of the reconstruction of images from the latent dimension 128 for the Fashion-MNIST dataset for various initializations in both the fixed and learnable setting.

### A.8.2  CIFAR-100

The trend seen in Table 3, where the learnable scenario convincingly has the least reconstruction error, is confirmed by the results for CIFAR-100, as seen in Table 6. Once again, this could be explained by the explicit training that the parameters of the encoder have undergone in the learnable scenario.

Similar to the results for CIFAR-10, we notice that the learnable scenario generally performs the best in terms of SSIM for CIFAR-100, as evidenced by Table 7, with the notable exception of $dim = 16$ where, once again, the fixed scenario demonstrates better performance, albeit by a small margin.

Table 8 further confirms some earlier observations, and reiterates that the images generated in the fixed scenario are more similar to the inputs than the ones generated by the learnable scenario. While the margins are not high, it is still noteworthy that the representations generated by the encoder had no task specific information. The only two exceptions here are $dim = 128$ and $dim = 256$, where the learnable scenario performs better.

### A.8.3 Fashion-MNIST

We further study the metrics in both these scenarios for the monochromatic images in the Fashion-MNIST dataset. Table 9 confirms that across all datasets, the lowest reconstruction losses are achieved by the learnable scenario where task specific representations are learnt with the primary objective of reducing the reconstruction loss.

Table 10 establishes that the best SSIM scores for this dataset are consistently achieved by the learnable scenario, confirming the trend so far as well. It is important to note that unlike the multi-channel images studied so far, there is no configuration of fixed weights that outperforms learnable weights.

Differing from the multi-channel image trend, we notice from Table 11 that for the monochromatic images in this dataset, the best FID scores are also achieved by the learnable scenario. This is different from the insights shown for multi-channel images.

In summary, it can be seen that learnable weights mostly achieve the best performance across all datasets and metrics, with the exception of FID for multi-channel RGB images. While the superior performance of learnable weights is not surprising, the comparability, margin of difference, and superior performance in FID for multi-channel images is an interesting finding. All these insights condense to the fact that while fixed random weights will never be as good as learnable weights, they are still able to generate reasonable embeddings, as evidenced in our closed setting experiments. This study further quantifies and qualifies the information captured by them and appeals for their use as a baseline at the very least.

Table 3: Reconstruction Loss (↓) for CIFAR-10

| Initialization Method | 16 | | 32 | | 64 | | 128 | | 256 | | 512 | |
|---|---|---|---|---|---|---|---|---|---|---|---|---|
| | Learnable | Fixed | Learnable | Fixed | Learnable | Fixed | Learnable | Fixed | Learnable | Fixed | Learnable | Fixed |
| Gaussian ($\mu = 0, \sigma = 0.02$) | 0.0169 | 0.0180 | 0.0120 | 0.0175 | 0.0080 | 0.0094 | 0.0049 | 0.0093 | 0.0148 | 0.0157 | 0.0618 | 0.0618 |
| Gaussian ($\mu = 0, \sigma = 0.5$) | 0.0168 | 0.0170 | **0.0117** | 0.0130 | **0.0079** | 0.0080 | **0.0048** | 0.0059 | 0.0034 | 0.0081 | **0.0080** | 0.0125 |
| Gaussian, ($\mu = 0, \sigma = 1$) | **0.0167** | 0.0170 | 0.0118 | 0.0131 | 0.0079 | 0.0080 | 0.0049 | 0.0060 | 0.0039 | 0.0085 | 0.0129 | 0.2961 |
| Gaussian ($\mu = 1, \sigma = 0.02$) | 0.0168 | 0.1700 | 0.0618 | 0.3005 | 0.0618 | 0.2926 | 0.0618 | 0.3068 | 0.0618 | 0.3081 | 0.0618 | 0.3058 |
| Gaussian ($\mu = 1, \sigma = 0.5$) | 0.0168 | 0.1669 | 0.0118 | 0.3018 | 0.0079 | 0.2921 | 0.0066 | 0.3067 | 0.0052 | 0.3058 | 0.0231 | 0.3036 |
| Gaussian ($\mu = 1, \sigma = 1$) | 0.0168 | 0.1660 | **0.0117** | 0.2998 | **0.0079** | 0.2928 | 0.0049 | 0.3047 | 0.0037 | 0.3148 | 0.0113 | 0.3062 |
| Gaussian ($\mu = -1, \sigma = 0.02$) | 0.0618 | 0.0618 | 0.0618 | 0.0618 | 0.0618 | 0.0618 | 0.0618 | 0.0618 | 0.0618 | 0.0618 | 0.0618 | 0.0618 |
| Gaussian ($\mu = -1, \sigma = 0.5$) | 0.0349 | 0.0618 | 0.0618 | 0.0618 | 0.0190 | 0.0616 | 0.0506 | 0.0617 | 0.0520 | 0.0617 | 0.0618 | 0.0618 |
| Gaussian ($\mu = -1, \sigma = 1$) | 0.0169 | 0.0594 | 0.0319 | 0.0608 | 0.0080 | 0.0571 | 0.0060 | 0.0606 | 0.0047 | 0.0606 | 0.0221 | 0.0616 |
| Orthogonal (Gain = 0.5) | 0.0168 | 0.0175 | 0.0119 | 0.0137 | 0.0080 | 0.0086 | **0.0048** | 0.0072 | 0.0032 | 0.0131 | 0.0091 | 0.0618 |
| Orthogonal (Gain = 1) | 0.0168 | 0.0172 | **0.0117** | 0.0132 | **0.0079** | 0.0084 | 0.0049 | 0.0062 | 0.0030 | 0.0083 | 0.0092 | 0.0130 |
| Orthogonal (Gain = 1.5) | 0.0168 | 0.0171 | **0.0117** | 0.0130 | **0.0079** | 0.0083 | 0.0049 | 0.0060 | 0.0031 | 0.0077 | 0.0085 | 0.0120 |
| Uniform, (-0.02, 0.02) | 0.0169 | 0.0201 | 0.0120 | 0.0618 | 0.0080 | 0.0101 | 0.0049 | 0.0343 | 0.0267 | 0.0618 | 0.0618 | 0.0618 |
| Uniform (-0.5, 0.5) | 0.0168 | 0.0172 | **0.0117** | 0.0132 | **0.0079** | 0.0081 | **0.0048** | 0.0058 | **0.0029** | 0.0074 | 0.0084 | 0.0116 |
| Uniform (-1, 1) | **0.0167** | 0.0171 | **0.0117** | 0.0132 | **0.0079** | 0.0080 | **0.0048** | 0.0058 | 0.0036 | 0.0078 | 0.0084 | 0.0123 |
| Uniform (-1.5, 1.5) | 0.0168 | 0.0171 | **0.0117** | 0.0132 | **0.0079** | 0.0080 | 0.0049 | 0.0059 | 0.0038 | 0.0080 | 0.0117 | 0.2377 |
| Uniform (0, 1) | 0.0168 | 0.0191 | 0.0119 | 0.2392 | **0.0079** | 0.2349 | 0.0055 | 0.3007 | 0.0528 | 0.3019 | 0.0418 | 0.3065 |
| Xavier Normal (Gain = 0.5) | 0.0169 | 0.0177 | 0.0119 | 0.0143 | 0.0080 | 0.0089 | **0.0048** | 0.0106 | 0.0035 | 0.0618 | 0.0100 | 0.0618 |
| Xavier Normal (Gain = 1) | 0.0168 | 0.0175 | 0.0118 | 0.0137 | **0.0079** | 0.0086 | **0.0048** | 0.0074 | 0.0035 | 0.0137 | 0.0303 | 0.0618 |
| Xavier Normal (Gain = 1.5) | 0.0168 | 0.0173 | **0.0117** | 0.0135 | 0.0080 | 0.0084 | **0.0048** | 0.0063 | 0.0032 | 0.0083 | 0.0085 | 0.0130 |
| Xavier Uniform (Gain = 0.5) | 0.0168 | 0.0177 | 0.0119 | 0.0194 | **0.0079** | 0.0090 | **0.0048** | 0.0586 | 0.0035 | 0.0618 | 0.0094 | 0.0618 |
| Xavier Uniform (Gain = 1) | 0.0168 | 0.0174 | 0.0119 | 0.0134 | **0.0079** | 0.0085 | **0.0048** | 0.0066 | 0.0033 | 0.0122 | 0.0097 | 0.0341 |
| Xavier Uniform (Gain = 1.5) | 0.0168 | 0.0173 | 0.0118 | 0.0130 | **0.0079** | 0.0084 | **0.0048** | 0.0061 | 0.0031 | 0.0082 | 0.0085 | 0.0139 |
| Kaiming Normal (Mode = fan_in) | 0.0168 | 0.0172 | **0.0117** | 0.0133 | **0.0079** | 0.0083 | 0.0049 | 0.0062 | 0.0031 | 0.0082 | 0.0082 | 0.0126 |
| Kaiming Normal (Mode = fan_out) | 0.0168 | 0.0174 | **0.0117** | 0.0137 | 0.0080 | 0.0085 | 0.0049 | 0.0065 | 0.0030 | 0.0093 | 0.0087 | 0.0144 |
| Kaiming Uniform (Mode = fan_in) | 0.0168 | 0.0172 | **0.0117** | 0.0132 | **0.0079** | 0.0083 | **0.0048** | 0.0061 | 0.0030 | 0.0078 | 0.0088 | 0.0 12 |
| Kaiming Uniform (Mode = fan_out) | 0.0168 | 0.0174 | 0.0119 | 0.0132 | **0.0079** | 0.0084 | **0.0048** | 0.0062 | 0.0034 | 0.0097 | 0.0086 | 0.0186 |
| Hadamard | 0.0168 | 0.0172 | 0.0118 | 0.0133 | **0.0079** | 0.0084 | **0.0048** | 0.0061 | 0.0031 | 0.0081 | 0.0084 | 0.0138 |

Table 4: SSIM (↑) for CIFAR-10

| Initialization Method | 16 | | 32 | | 64 | | 128 | | 256 | | 512 | |
|---|---|---|---|---|---|---|---|---|---|---|---|---|
| | Learnable | Fixed | Learnable | Fixed | Learnable | Fixed | Learnable | Fixed | Learnable | Fixed | Learnable | Fixed |
| Gaussian ($\mu=0,\sigma=0.02$) | 0.3818 | 0.3688 | 0.5091 | 0.4671 | 0.6449 | 0.6291 | 0.7761 | 0.6400 | 0.7060 | 0.4673 | 0.1323 | 0.1323 |
| Gaussian ($\mu=0,\sigma=0.5$) | 0.3813 | 0.3788 | 0.5049 | 0.4874 | **0.6501** | 0.6444 | 0.7764 | 0.7446 | 0.8344 | 0.6854 | **0.6414** | 0.5553 |
| Gaussian ($\mu=0,\sigma=1$) | 0.3797 | 0.3763 | 0.5052 | 0.4857 | 0.6487 | 0.6422 | 0.7749 | 0.7410 | 0.8135 | 0.6708 | 0.4798 | 0.0038 |
| Gaussian ($\mu=1,\sigma=0.02$) | 0.3822 | 0.1213 | 0.1323 | 0.0038 | 0.1324 | 0.0031 | 0.1324 | 0.0037 | 0.1323 | 0.0037 | 0.1323 | 0.0035 |
| Gaussian ($\mu=1,\sigma=0.5$) | 0.3820 | 0.1206 | 0.5070 | 0.0037 | 0.6478 | 0.0026 | 0.7094 | 0.0037 | 0.7572 | 0.0034 | 0.3989 | 0.0037 |
| Gaussian ($\mu=1,\sigma=1$) | 0.3819 | 0.1214 | 0.5048 | 0.0035 | 0.6470 | 0.0032 | 0.7747 | 0.0037 | 0.8209 | 0.0038 | 0.5222 | 0.0032 |
| Gaussian ($\mu=-1,\sigma=0.02$) | 0.1324 | 0.1323 | 0.1322 | 0.1322 | 0.1323 | 0.1323 | 0.1323 | 0.1323 | 0.1323 | 0.1323 | 0.1323 | 0.1323 |
| Gaussian ($\mu=-1,\sigma=0.5$) | 0.2819 | 0.1324 | 0.1323 | 0.1322 | 0.5431 | 0.1326 | 0.2502 | 0.1324 | 0.2041 | 0.1324 | 0.1323 | 0.1323 |
| Gaussian ($\mu=-1,\sigma=1$) | 0.3811 | 0.1383 | 0.3574 | 0.1338 | 0.6466 | 0.1460 | 0.7303 | 0.1341 | 0.7765 | 0.1339 | 0.4189 | 0.1324 |
| Orthogonal (Gain = 0.5) | 0.3822 | 0.3753 | 0.5084 | 0.4841 | 0.6448 | 0.6326 | 0.7758 | 0.7051 | 0.8455 | 0.5130 | 0.5972 | 0.5130 |
| Orthogonal (Gain = 1) | 0.3821 | 0.3787 | 0.5047 | 0.4902 | 0.6465 | 0.6367 | 0.7754 | 0.7353 | 0.8523 | 0.6686 | 0.5962 | 0.5210 |
| Orthogonal (Gain = 1.5) | 0.3819 | 0.3797 | 0.5055 | 0.4887 | 0.6471 | 0.6396 | 0.7760 | 0.7402 | 0.8503 | 0.6906 | 0.6230 | 0.5565 |
| Uniform (-0.02, 0.02) | 0.3814 | 0.3686 | 0.5088 | 0.1322 | 0.6453 | 0.6111 | 0.7758 | 0.3554 | 0.5552 | 0.1323 | 0.1323 | 0.1323 |
| Uniform (-0.5, 0.5) | 0.3809 | 0.3778 | 0.5053 | 0.4826 | 0.6488 | 0.6432 | **0.7774** | 0.7492 | **0.8559** | 0.7107 | 0.6246 | 0.5780 |
| Uniform (-1, 1) | 0.3794 | 0.3766 | 0.5044 | 0.4837 | 0.6493 | 0.6460 | 0.7761 | 0.7493 | 0.8285 | 0.6905 | 0.6246 | 0.5528 |
| Uniform (-1.5, 1.5) | 0.3806 | 0.3764 | 0.5046 | 0.4821 | 0.6487 | 0.6437 | 0.7754 | 0.7464 | 0.8178 | 0.6881 | 0.5048 | 0.1062 |
| Uniform (0, 1) | 0.3823 | **0.3883** | 0.5074 | 0.0850 | 0.6476 | 0.1227 | 0.7474 | 0.0038 | 0.1814 | 0.0036 | 0.2872 | 0.0039 |
| Xavier Normal (Gain = 0.5) | 0.3819 | 0.3728 | 0.5084 | 0.4770 | 0.6449 | 0.6294 | 0.7760 | 0.5941 | 0.8293 | 0.1324 | 0.5667 | 0.1324 |
| Xavier Normal (Gain = 1) | 0.3820 | 0.3759 | 0.5086 | 0.4817 | 0.6467 | 0.6310 | 0.7764 | 0.6927 | 0.8310 | 0.5015 | 0.4051 | 0.1324 |
| Xavier Normal (Gain = 1.5) | 0.3821 | 0.3773 | 0.5052 | 0.4849 | 0.6461 | 0.6364 | 0.7763 | 0.7336 | 0.8447 | 0.6747 | 0.6199 | 0.5261 |
| Xavier Uniform (Gain = 0.5) | 0.3819 | 0.3738 | 0.5103 | 0.3766 | 0.6469 | 0.6213 | 0.7762 | 0.1334 | 0.8286 | 0.1324 | 0.5880 | 0.1324 |
| Xavier Uniform (Gain = 1) | 0.3817 | 0.3762 | 0.5106 | 0.4879 | 0.6470 | 0.6349 | 0.7762 | 0.7222 | 0.8384 | 0.5436 | 0.5775 | 0.2318 |
| Xavier Uniform (Gain = 1.5) | 0.3819 | 0.3773 | 0.5076 | 0.4858 | 0.6474 | 0.6368 | 0.7762 | 0.7412 | 0.8490 | 0.6739 | 0.6216 | 0.4869 |
| Kaiming Normal (Mode = fan_in) | 0.3823 | 0.3776 | 0.5044 | 0.4840 | 0.6471 | 0.6384 | 0.7760 | 0.7361 | 0.8502 | 0.6811 | 0.6331 | 0.5473 |
| Kaiming Normal (Mode = fan_out) | 0.3820 | 0.3760 | 0.5052 | 0.4847 | 0.6461 | 0.6346 | 0.7757 | 0.7275 | 0.8544 | 0.6499 | 0.6130 | 0.4864 |
| Kaiming Uniform (Mode = fan_in) | 0.3818 | 0.3777 | 0.5038 | 0.4828 | 0.6482 | 0.6399 | 0.7762 | 0.7410 | 0.8528 | 0.6928 | 0.6103 | 0.5482 |
| Kaiming Uniform (Mode = fan_out) | 0.3820 | 0.3765 | **0.5108** | 0.4900 | 0.6467 | 0.6362 | 0.7761 | 0.7383 | 0.8364 | 0.6211 | 0.6164 | 0.3749 |
| Hadamard | 0.3817 | 0.3778 | 0.5045 | 0.4896 | 0.6474 | 0.6402 | 0.7764 | 0.7461 | 0.8501 | 0.6827 | 0.6264 | 0.4918 |

Table 5: FID (↓) for CIFAR-10

| Initialization Method | 16 | | 32 | | 64 | | 128 | | 256 | | 512 | |
|---|---|---|---|---|---|---|---|---|---|---|---|---|
| | Learnable | Fixed | Learnable | Fixed | Learnable | Fixed | Learnable | Fixed | Learnable | Fixed | Learnable | Fixed |
| Gaussian ($\mu=0,\sigma=0.02$) | 203.8714 | 245.8416 | 170.1749 | 241.8876 | 132.6609 | 165.0251 | 99.7668 | 161.4639 | 145.1659 | 222.6520 | 456.7696 | 456.7786 |
| Gaussian ($\mu=0,\sigma=0.5$) | 200.8313 | 197.7794 | 166.7423 | 161.6618 | 128.1037 | 124.6547 | 97.9044 | 97.7828 | 72.7366 | 96.9689 | 138.9144 | 123.1540 |
| Gaussian ($\mu=0,\sigma=1$) | 197.5735 | **193.1815** | 167.9353 | **160.4486** | 124.1292 | 120.0340 | 97.0217 | 94.0685 | 81.0270 | 97.6529 | 182.4072 | 478.5343 |
| Gaussian ($\mu=1,\sigma=0.02$) | 198.9519 | 197.7125 | 458.4298 | 515.2014 | 467.9735 | 467.8844 | 447.5934 | 516.8441 | 453.9875 | 490.4471 | 451.0224 | 459.6968 |
| Gaussian ($\mu=1,\sigma=0.5$) | 198.5620 | 217.2923 | 167.4394 | 504.0632 | 129.1247 | 488.1917 | 115.5052 | 473.5292 | 102.6466 | 476.8645 | 241.3645 | 481.8716 |
| Gaussian ($\mu=1,\sigma=1$) | 197.9122 | 217.5943 | 167.3174 | 497.2559 | 131.6604 | 507.0174 | 101.5437 | 505.2754 | 78.6406 | 473.9390 | 169.2239 | 541.3786 |
| Gaussian ($\mu=-1,\sigma=0.02$) | 453.9857 | 457.6421 | 461.8598 | 461.5971 | 447.4334 | 446.9508 | 453.2713 | 454.0710 | 455.1198 | 455.1721 | 454.6053 | 457.0475 |
| Gaussian ($\mu=-1,\sigma=0.5$) | 300.1751 | 456.1936 | 454.9631 | 456.8790 | 200.3819 | 450.1558 | 385.8363 | 445.2631 | 404.6956 | 448.4085 | 458.4824 | 454.3195 |
| Gaussian ($\mu=-1,\sigma=1$) | 211.8576 | 417.1591 | 290.7107 | 428.3455 | 136.8156 | 376.5017 | 113.8027 | 420.2316 | 94.7660 | 440.8685 | 232.7105 | 451.4181 |
| Orthogonal (Gain = 0.5) | 202.5649 | 234.6800 | 167.1068 | 202.1423 | 132.7072 | 149.2592 | 99.1106 | 123.7785 | 67.8194 | 192.8911 | 152.4817 | 456.7346 |
| Orthogonal (Gain = 1) | 204.5313 | 223.9168 | 167.0671 | 193.3545 | 131.4684 | 148.1803 | 101.9759 | 109.1293 | 65.3211 | 119.5430 | 152.5085 | 173.6032 |
| Orthogonal (Gain = 1.5) | 201.0340 | 215.1335 | 167.5513 | 183.8371 | 133.0137 | 144.4053 | 101.6463 | 108.1394 | 66.2897 | 110.1505 | 145.8183 | 152.0322 |
| Uniform (-0.02, 0.02) | 204.5676 | 262.6497 | 169.9517 | 461.5289 | 131.1087 | 180.3531 | 100.0333 | 326.2196 | 229.5681 | 454.0649 | 452.6827 | 460.8555 |
| Uniform (-0.5, 0.5) | 200.0021 | 209.9359 | 166.1523 | 170.8800 | 130.0720 | 133.0439 | 98.1676 | 98.4345 | **63.0506** | 91.2329 | 144.3277 | **121.5707** |
| Uniform (-1, 1) | 198.1587 | 197.9735 | 166.9635 | 161.8566 | 127.9312 | 123.3643 | 98.9669 | 94.7042 | 75.6332 | 94.0704 | 143.2481 | 124.3923 |
| Uniform (-1.5, 1.5) | 200.1311 | 198.2672 | 166.7834 | 163.4821 | 126.1630 | **119.6831** | 97.2993 | **93.7838** | 79.0227 | 93.6728 | 173.1077 | 432.3132 |
| Uniform (0, 1) | 203.5686 | 218.0428 | 170.7694 | 455.7111 | 131.8741 | 452.5606 | 106.3817 | 487.4816 | 395.4356 | 514.1393 | 346.4521 | 471.4219 |
| Xavier Normal (Gain = 0.5) | 203.4820 | 235.8485 | 166.6828 | 207.8234 | 132.5345 | 152.1559 | 98.8247 | 175.5650 | 73.7299 | 457.8352 | 158.3653 | 449.4362 |
| Xavier Normal (Gain = 1) | 201.7016 | 230.3907 | 166.2114 | 194.1856 | 131.0969 | 147.9112 | 97.8332 | 122.9046 | 74.0737 | 196.6364 | 271.9412 | 457.4161 |
| Xavier Normal (Gain = 1.5) | 202.1547 | 227.7135 | 166.0910 | 192.1118 | 131.3674 | 148.8128 | 100.3043 | 109.6404 | 68.4997 | 119.3479 | 146.8924 | 171.5187 |
| Xavier Uniform (Gain = 0.5) | 203.0753 | 236.7116 | 166.8513 | 248.0841 | 133.0265 | 152.9370 | 97.9908 | 445.9062 | 75.0038 | 452.1075 | 153.8430 | 452.7558 |
| Xavier Uniform (Gain = 1) | 203.8311 | 234.0839 | 165.6804 | 198.8899 | 132.4623 | 148.5412 | 98.2993 | 113.5351 | 70.9328 | 179.9320 | 156.3881 | 325.7304 |
| Xavier Uniform (Gain = 1.5) | 202.4615 | 229.3706 | 166.4272 | 188.8543 | 131.1432 | 148.3616 | 99.2122 | 108.1039 | 66.3070 | 121.7508 | 146.7658 | 182.7901 |
| Kaiming Normal (Mode = fan_in) | 203.0479 | 218.8304 | 171.2693 | 186.5371 | 134.1824 | 145.4180 | 101.8193 | 109.3164 | 66.4424 | 113.3181 | 142.9782 | 154.7894 |
| Kaiming Normal (Mode = fan_out) | 205.2424 | 231.6536 | 167.8388 | 198.4719 | 131.0087 | 149.4044 | 101.1184 | 110.7348 | 64.0376 | 138.8431 | 148.8014 | 194.1676 |
| Kaiming Uniform (Mode = fan_in) | 201.3271 | 222.4708 | 167.4424 | 186.3614 | 133.1087 | 145.1121 | 100.5890 | 107.6714 | 64.7305 | 111.1232 | 149.3870 | 155.9740 |
| Kaiming Uniform (Mode = fan_out) | 202.2209 | 232.4030 | 166.1359 | 197.0349 | 130.6100 | 148.7798 | 98.9740 | 108.4047 | 71.9651 | 147.0019 | 148.0916 | 230.0073 |
| Hadamard | 200.8951 | 227.0174 | 167.8796 | 193.8820 | 133.3731 | 148.5855 | 99.5843 | 107.3967 | 66.2025 | 120.1903 | 145.5386 | 184.9573 |

## Table 6: Reconstruction Loss (↓) for CIFAR-100

| Initialization Method | 16 | | 32 | | 64 | | 128 | | 256 | | 512 | |
|---|---|---|---|---|---|---|---|---|---|---|---|---|
| | Learnable | Fixed | Learnable | Fixed | Learnable | Fixed | Learnable | Fixed | Learnable | Fixed | Learnable | Fixed |
| Gaussian ($\mu=0, \sigma=0.02$) | 0.0171 | 0.0196 | 0.0118 | 0.0176 | 0.0081 | 0.0099 | 0.0050 | 0.0101 | 0.0035 | 0.0141 | 0.0705 | 0.0705 |
| Gaussian ($\mu=0, \sigma=0.5$) | 0.0170 | 0.0173 | **0.0117** | 0.0133 | **0.0080** | 0.0082 | 0.0050 | 0.0061 | 0.0038 | 0.0084 | **0.0085** | 0.0130 |
| Gaussian ($\mu=0, \sigma=1$) | **0.0169** | 0.0173 | 0.0118 | 0.0134 | **0.0080** | 0.0082 | 0.0050 | 0.0063 | 0.0039 | 0.0087 | 0.0232 | 0.2460 |
| Gaussian ($\mu=1, \sigma=0.02$) | 0.0170 | 0.1709 | 0.0705 | 0.3072 | 0.0334 | 0.3006 | 0.0705 | 0.3127 | 0.0705 | 0.3171 | 0.0705 | 0.3085 |
| Gaussian ($\mu=1, \sigma=0.5$) | 0.0171 | 0.1693 | 0.0119 | 0.3029 | **0.0080** | 0.2431 | 0.0055 | 0.3153 | 0.0177 | 0.3134 | 0.0120 | 0.3114 |
| Gaussian ($\mu=1, \sigma=1$) | 0.0172 | 0.1713 | **0.0117** | 0.3063 | 0.0081 | 0.3004 | 0.0050 | 0.3169 | 0.0040 | 0.3158 | 0.0107 | 0.3176 |
| Gaussian ($\mu=-1, \sigma=0.02$) | 0.0705 | 0.0705 | 0.0705 | 0.0705 | 0.0705 | 0.0705 | 0.0705 | 0.0705 | 0.0705 | 0.0705 | 0.0705 | 0.0705 |
| Gaussian ($\mu=-1, \sigma=0.5$) | 0.0280 | 0.0705 | 0.0705 | 0.0705 | 0.0208 | 0.0700 | 0.0197 | 0.0703 | 0.0488 | 0.0703 | 0.0603 | 0.0705 |
| Gaussian ($\mu=-1, \sigma=1$) | 0.0172 | 0.0670 | 0.0354 | 0.0690 | 0.0081 | 0.0642 | 0.0050 | 0.0685 | 0.0042 | 0.0685 | 0.0114 | 0.0702 |
| Orthogonal (Gain = 0.5) | 0.0170 | 0.0180 | 0.0119 | 0.0143 | 0.0081 | 0.0088 | 0.0050 | 0.0075 | 0.0034 | 0.0140 | 0.0093 | 0.0705 |
| Orthogonal (Gain = 1) | **0.0169** | 0.0174 | 0.0118 | 0.0137 | 0.0081 | 0.0085 | 0.0050 | 0.0064 | 0.0034 | 0.0086 | 0.0097 | 0.0135 |
| Orthogonal (Gain = 1.5) | **0.0169** | 0.0173 | **0.0117** | 0.0132 | 0.0081 | 0.0084 | 0.0050 | 0.0062 | 0.0034 | 0.0080 | 0.0097 | 0.0124 |
| Uniform (-0.02, 0.02) | 0.0171 | 0.0224 | 0.0121 | 0.0705 | 0.0081 | 0.0108 | 0.0050 | 0.0195 | 0.0304 | 0.0705 | 0.0705 | 0.0705 |
| Uniform (-0.5, 0.5) | **0.0169** | 0.0174 | **0.0117** | 0.0135 | **0.0080** | 0.0083 | **0.0049** | 0.0060 | **0.0033** | 0.0077 | 0.0087 | 0.0122 |
| Uniform (-1 to 1) | **0.0169** | 0.0174 | **0.0117** | 0.0135 | **0.0080** | 0.0082 | 0.0050 | 0.0060 | 0.0038 | 0.0081 | 0.0089 | 0.0131 |
| Uniform (-1.5, 1.5) | **0.0169** | 0.0174 | 0.0118 | 0.0135 | **0.0080** | 0.0082 | 0.0050 | 0.0061 | 0.0038 | 0.0084 | 0.0111 | 0.3030 |
| Uniform (0, 1) | 0.0171 | 0.0202 | 0.0118 | 0.3012 | **0.0080** | 0.2987 | 0.0056 | 0.3141 | 0.0588 | 0.3182 | 0.0466 | 0.3133 |
| Xavier Normal (Gain = 0.5) | 0.0171 | 0.0183 | 0.0118 | 0.0151 | 0.0081 | 0.0091 | 0.0050 | 0.0114 | 0.0036 | 0.0705 | 0.0095 | 0.0705 |
| Xavier Normal (Gain = 1) | 0.0170 | 0.0179 | **0.0117** | 0.0141 | 0.0081 | 0.0088 | 0.0050 | 0.0076 | 0.0036 | 0.0142 | 0.0098 | 0.0628 |
| Xavier Normal (Gain = 1.5) | 0.0170 | 0.0176 | **0.0117** | 0.0139 | 0.0081 | 0.0086 | 0.0050 | 0.0066 | 0.0035 | 0.0087 | 0.0095 | 0.0135 |
| Xavier Uniform (Gain = 0.5) | 0.0172 | 0.0183 | 0.0118 | 0.0168 | 0.0081 | 0.0093 | 0.0050 | 0.0274 | 0.0037 | 0.0705 | 0.0107 | 0.0705 |
| Xavier Uniform (Gain = 1) | 0.0170 | 0.0178 | 0.0118 | 0.0137 | 0.0081 | 0.0087 | 0.0050 | 0.0071 | 0.0037 | 0.0126 | 0.0090 | 0.0454 |
| Xavier Uniform (Gain = 1.5) | **0.0169** | 0.0176 | 0.0118 | 0.0133 | 0.0081 | 0.0086 | 0.0050 | 0.0064 | 0.0035 | 0.0085 | 0.0097 | 0.0144 |
| Kaiming Normal (Mode = fan_in) | **0.0169** | 0.0175 | 0.0118 | 0.0136 | 0.0081 | 0.0085 | 0.0050 | 0.0065 | **0.0033** | 0.0086 | 0.0098 | 0.0131 |
| Kaiming Normal (Mode = fan_out) | **0.0169** | 0.0179 | **0.0117** | 0.0142 | 0.0081 | 0.0088 | 0.0050 | 0.0068 | 0.0034 | 0.0095 | 0.0095 | 0.0149 |
| Kaiming Uniform (Mode = fan_in) | **0.0169** | 0.0174 | **0.0117** | 0.0134 | **0.0080** | 0.0084 | 0.0050 | 0.0064 | 0.0034 | 0.0081 | 0.0096 | 0.0129 |
| Kaiming Uniform (Mode = fan_out) | **0.0169** | 0.0176 | 0.0118 | 0.0134 | 0.0081 | 0.0086 | 0.0050 | 0.0065 | 0.0036 | 0.0099 | 0.0097 | 0.0192 |
| Hadamard | **0.0169** | 0.0174 | 0.0118 | 0.0136 | 0.0081 | 0.0085 | 0.0050 | 0.0063 | 0.0035 | 0.0086 | 0.0095 | 0.0149 |

## Table 7: SSIM (↑) for CIFAR-100

| Initialization Method | 16 | | 32 | | 64 | | 128 | | 256 | | 512 | |
|---|---|---|---|---|---|---|---|---|---|---|---|---|
| | Learnable | Fixed | Learnable | Fixed | Learnable | Fixed | Learnable | Fixed | Learnable | Fixed | Learnable | Fixed |
| Gaussian ($\mu=0, \sigma=0.02$) | 0.3757 | 0.3721 | 0.4959 | 0.4578 | 0.6263 | 0.6092 | 0.7591 | 0.6198 | 0.8241 | 0.4982 | 0.1413 | 0.1413 |
| Gaussian ($\mu=0, \sigma=0.5$) | 0.3745 | 0.3665 | 0.4976 | 0.4793 | **0.6312** | 0.6234 | 0.7583 | 0.7255 | 0.8095 | 0.6644 | **0.6104** | 0.5328 |
| Gaussian ($\mu=0, \sigma=1$) | 0.3745 | 0.3667 | 0.4960 | 0.4769 | 0.6293 | 0.6221 | 0.7569 | 0.7214 | 0.8028 | 0.6554 | 0.4416 | 0.1033 |
| Gaussian ($\mu=1, \sigma=0.02$) | 0.3751 | 0.1154 | 0.1413 | 0.0038 | 0.1413 | 0.0036 | 0.1412 | 0.0038 | 0.1413 | 0.0035 | 0.1413 | 0.0035 |
| Gaussian ($\mu=1, \sigma=0.5$) | 0.3766 | 0.1090 | **0.4977** | 0.0035 | 0.6300 | 0.1183 | 0.7367 | 0.0034 | 0.6505 | 0.0111 | 0.4980 | 0.0034 |
| Gaussian ($\mu=1, \sigma=1$) | 0.3775 | 0.1136 | 0.4967 | 0.0039 | 0.6295 | 0.0034 | 0.7564 | 0.0035 | 0.8015 | 0.0033 | 0.5323 | 0.0033 |
| Gaussian ($\mu=-1, \sigma=0.02$) | 0.1413 | 0.1414 | 0.1413 | 0.1413 | 0.1413 | 0.1413 | 0.1412 | 0.1412 | 0.1413 | 0.1413 | 0.1414 | 0.1414 |
| Gaussian ($\mu=-1, \sigma=0.5$) | 0.3304 | 0.1414 | 0.1414 | 0.1414 | 0.5307 | 0.1418 | 0.5663 | 0.1414 | 0.2482 | 0.1414 | 0.1807 | 0.1413 |
| Gaussian ($\mu=-1, \sigma=1$) | 0.3767 | 0.1478 | 0.3535 | 0.1432 | 0.6287 | 0.1551 | 0.7576 | 0.1438 | 0.7899 | 0.1434 | 0.5081 | 0.1416 |
| Orthogonal (Gain = 0.5) | 0.3752 | 0.3722 | 0.4968 | 0.4776 | 0.6272 | 0.6166 | 0.7589 | 0.6837 | 0.8264 | 0.4968 | 0.5792 | 0.1414 |
| Orthogonal (Gain = 1) | 0.3729 | 0.3671 | 0.4958 | 0.4838 | 0.6284 | 0.6215 | 0.7592 | 0.7171 | 0.8294 | 0.6518 | 0.5661 | 0.5039 |
| Orthogonal (Gain = 1.5) | 0.3742 | 0.3687 | 0.4966 | 0.4821 | 0.6303 | 0.6225 | 0.7599 | 0.7218 | 0.8284 | 0.6717 | 0.5650 | 0.5418 |
| Uniform (-0.02, 0.02) | 0.3764 | 0.3699 | 0.4976 | 0.1413 | 0.6271 | 0.5948 | 0.7583 | 0.4727 | 0.5451 | 0.1413 | 0.1413 | 0.1414 |
| Uniform (-0.5, 0.5) | 0.3742 | 0.3653 | **0.4977** | 0.4753 | 0.6311 | 0.6232 | **0.7601** | 0.7294 | **0.8330** | 0.6884 | 0.6049 | 0.5543 |
| Uniform (-1, 1) | 0.3741 | 0.3666 | 0.4962 | 0.4739 | 0.6309 | 0.6245 | 0.7582 | 0.7288 | 0.8082 | 0.6720 | 0.5946 | 0.5244 |
| Uniform (-1.5, 1.5) | 0.3740 | 0.3670 | 0.4964 | 0.4726 | 0.6297 | 0.6232 | 0.7574 | 0.7273 | 0.8063 | 0.6644 | 0.5184 | 0.0074 |
| Uniform (0, 1) | 0.3764 | **0.3842** | 0.4965 | 0.0033 | 0.6305 | 0.0041 | 0.7297 | 0.0038 | 0.2148 | 0.0034 | 0.2963 | 0.0110 |
| Xavier Normal (Gain = 0.5) | 0.3762 | 0.3745 | 0.4962 | 0.4673 | 0.6270 | 0.6121 | 0.7582 | 0.5732 | 0.8187 | 0.1414 | 0.5748 | 0.1413 |
| Xavier Normal (Gain = 1) | 0.3738 | 0.3708 | 0.4964 | 0.4737 | 0.6282 | 0.6133 | 0.7584 | 0.6742 | 0.8156 | 0.4877 | 0.5624 | 0.1608 |
| Xavier Normal (Gain = 1.5) | 0.3744 | 0.3670 | 0.4963 | 0.4769 | 0.6297 | 0.6196 | 0.7597 | 0.7150 | 0.8244 | 0.6526 | 0.5727 | 0.5088 |
| Xavier Uniform (Gain = 0.5) | 0.3777 | 0.3731 | 0.4954 | 0.4269 | 0.6283 | 0.6057 | 0.7579 | 0.3702 | 0.8111 | 0.1414 | 0.5387 | 0.1414 |
| Xavier Uniform (Gain = 1) | 0.3713 | 0.3702 | 0.4962 | 0.4783 | 0.6286 | 0.6181 | 0.7585 | 0.6969 | 0.8133 | 0.5272 | 0.5909 | 0.2173 |
| Xavier Uniform (Gain = 1.5) | 0.3733 | 0.3678 | 0.4963 | 0.4802 | 0.6289 | 0.6195 | 0.7586 | 0.7212 | 0.8230 | 0.6520 | 0.5656 | 0.4735 |
| Kaiming Normal (Mode = fan_in) | 0.3720 | 0.3664 | 0.4964 | 0.4771 | 0.6297 | 0.6219 | 0.7597 | 0.7171 | 0.8305 | 0.6613 | 0.5606 | 0.5298 |
| Kaiming Normal (Mode = fan_out) | 0.3730 | 0.3719 | 0.4966 | 0.4777 | 0.6287 | 0.6177 | 0.7592 | 0.7089 | 0.8284 | 0.6291 | 0.5717 | 0.4729 |
| Kaiming Uniform (Mode = fan_in) | 0.3740 | 0.3656 | 0.4973 | 0.4762 | 0.6309 | 0.6213 | 0.7594 | 0.7219 | 0.8266 | 0.6736 | 0.5689 | 0.5273 |
| Kaiming Uniform (Mode = fan_out) | 0.3723 | 0.3697 | 0.4965 | 0.4810 | 0.6288 | 0.6187 | 0.7583 | 0.7191 | 0.8180 | 0.6039 | 0.5674 | 0.3705 |
| Hadamard | 0.3723 | 0.3689 | 0.4960 | 0.4800 | 0.6293 | 0.6223 | 0.7597 | 0.7272 | 0.8241 | 0.6594 | 0.5724 | 0.4652 |

Table 8: FID ($\downarrow$) for CIFAR-100

| Initialization Method | 16 | | 32 | | 64 | | 128 | | 256 | | 512 | |
|---|---|---|---|---|---|---|---|---|---|---|---|---|
| | Learnable | Fixed | Learnable | Fixed | Learnable | Fixed | Learnable | Fixed | Learnable | Fixed | Learnable | Fixed |
| Gaussian ($\mu = 0, \sigma = 0.02$) | 183.0714 | 226.4144 | 152.4853 | 218.1764 | 118.7887 | 151.8834 | 88.6712 | 155.8850 | 62.5579 | 197.5636 | 446.9524 | 453.7532 |
| Gaussian ($\mu = 0, \sigma = 0.5$) | 179.0554 | 177.1442 | 153.0044 | **149.3330** | 115.1627 | 113.5859 | 87.7116 | 91.1229 | 67.8090 | 99.8157 | 128.6850 | 123.7941 |
| Gaussian ($\mu = 0, \sigma = 1$) | 177.5401 | **173.9579** | 152.1853 | 150.7130 | 114.1189 | **109.7487** | **85.7010** | 89.7494 | 71.2509 | 98.5784 | 214.9221 | 425.7336 |
| Gaussian ($\mu = 1, \sigma = 0.02$) | 177.1480 | 193.7569 | 447.8329 | 524.0421 | 261.2349 | 509.9010 | 443.4188 | 500.5533 | 447.2748 | 507.4375 | 445.5235 | 476.8426 |
| Gaussian ($\mu = 1, \sigma = 0.5$) | 177.2612 | 213.6498 | 153.8454 | 519.6964 | 115.2407 | 463.5438 | 91.9068 | 473.3522 | 155.3197 | 480.8309 | 158.4461 | 486.2239 |
| Gaussian ($\mu = 1, \sigma = 1$) | 178.4474 | 194.9128 | 152.3325 | 480.8134 | 116.5051 | 534.0475 | 89.0951 | 495.8878 | 71.5585 | 461.1200 | 149.3136 | 474.6477 |
| Gaussian ($\mu = -1, \sigma = 0.02$) | 453.8588 | 453.3746 | 447.8728 | 443.8270 | 450.8869 | 449.4509 | 444.7023 | 447.1397 | 443.6439 | 446.9514 | 447.1573 | 446.8599 |
| Gaussian ($\mu = -1, \sigma = 0.5$) | 243.5210 | 455.3725 | 451.3300 | 447.1603 | 188.3488 | 448.6424 | 177.2609 | 441.8264 | 340.6086 | 442.2053 | 406.6731 | 449.2385 |
| Gaussian ($\mu = -1, \sigma = 1$) | 187.7530 | 406.3138 | 277.8171 | 419.5802 | 118.1635 | 364.8020 | 89.7250 | 424.6603 | 74.7700 | 438.4925 | 155.2045 | 434.9560 |
| Orthogonal (Gain = 0.5) | 181.9317 | 205.3326 | 153.4782 | 188.3867 | 118.8899 | 134.6475 | 88.6314 | 112.1980 | 61.8051 | 184.6875 | 138.1685 | 452.5748 |
| Orthogonal (Gain = 1) | 179.8894 | 192.3198 | 156.6029 | 176.4457 | 118.3627 | 127.3099 | 89.2277 | 98.0656 | 60.3823 | 112.6503 | 141.7588 | 162.5084 |
| Orthogonal (Gain = 1.5) | 180.5597 | 185.0709 | 153.4874 | 166.4058 | 117.6520 | 124.5160 | 88.7658 | 96.7367 | 60.3933 | 103.9144 | 140.9319 | 142.3769 |
| Uniform (-0.02, 0.02) | 181.4915 | 246.5185 | 158.3255 | 447.4950 | 117.4418 | 168.3182 | 89.0386 | 238.4189 | 220.0897 | 443.9430 | 450.3622 | 446.4369 |
| Uniform (-0.5, 0.5) | 179.1902 | 184.6318 | 151.2621 | 157.7564 | 116.1361 | 117.4907 | 87.8410 | 92.4923 | **58.6401** | 92.2858 | 131.2850 | **121.5421** |
| Uniform (-1, 1) | 178.1346 | 176.3214 | 152.5265 | 151.7006 | 114.9230 | 111.9121 | 87.4961 | 89.7354 | 68.1323 | 97.4910 | 133.0328 | 128.8066 |
| Uniform (-1.5, 1.5) | 177.6370 | 176.8045 | 153.1880 | 151.7605 | 113.6461 | 111.1578 | 86.8068 | 89.3090 | 68.8900 | 96.9063 | 154.0161 | 492.5122 |
| Uniform (0, 1) | 183.2623 | 201.6881 | 152.9768 | 539.8026 | 116.2003 | 541.1843 | 93.9946 | 541.0256 | 390.0802 | 511.8127 | 333.2009 | 520.5569 |
| Xavier Normal (Gain = 0.5) | 182.8722 | 210.3721 | 152.4383 | 192.9956 | 117.6267 | 137.6916 | 88.7232 | 167.4810 | 65.0197 | 456.4190 | 140.1085 | 452.3163 |
| Xavier Normal (Gain = 1) | 180.7308 | 201.6339 | 151.9775 | 178.9803 | 116.1575 | 132.4176 | 88.6833 | 111.9473 | 66.0399 | 177.6619 | 142.5526 | 424.4128 |
| Xavier Normal (Gain = 1.5) | 179.8782 | 193.4091 | 153.7713 | 176.1558 | 116.8737 | 128.5625 | 89.1732 | 98.4813 | 62.5334 | 111.8037 | 140.2330 | 161.7399 |
| Xavier Uniform (Gain = 0.5) | 184.2523 | 210.1855 | 152.8523 | 208.7189 | 117.7331 | 139.5411 | 88.0695 | 265.0084 | 67.7077 | 457.6977 | 148.6311 | 453.5656 |
| Xavier Uniform (Gain = 1) | 179.9830 | 201.9290 | 152.6019 | 180.2479 | 115.8996 | 131.9550 | 88.7228 | 105.3162 | 67.0680 | 165.5826 | 136.6790 | 344.7588 |
| Xavier Uniform (Gain = 1.5) | 179.5444 | 197.6550 | 153.3699 | 172.5248 | 117.2454 | 128.3960 | 88.9577 | 97.2003 | 63.3121 | 113.9055 | 141.5417 | 169.5280 |
| Kaiming Normal (Mode = fan_in) | 179.4043 | 187.2206 | 156.6531 | 166.1056 | 117.7342 | 125.3071 | 89.1772 | 98.3341 | 59.6919 | 107.2348 | 141.7981 | 145.5695 |
| Kaiming Normal (Mode = fan_out) | 179.0123 | 202.5186 | 151.8673 | 181.2652 | 117.1243 | 131.0979 | 89.1713 | 99.7737 | 60.9441 | 126.1473 | 139.8583 | 176.6325 |
| Kaiming Uniform (Mode = fan_in) | 179.4043 | 187.2206 | 156.6531 | 166.1056 | 117.7342 | 125.3071 | 89.1772 | 98.3341 | 59.6919 | 107.2348 | 141.7981 | 145.5695 |
| Kaiming Uniform (Mode = fan_out) | 179.0123 | 202.5186 | 151.8673 | 181.2652 | 117.1243 | 131.0979 | 89.1713 | 99.7737 | 60.9441 | 126.1473 | 139.8583 | 176.6325 |
| Hadamard | 179.9593 | 193.9618 | 153.6315 | 174.6738 | 116.8275 | 127.4908 | 88.3731 | 95.7952 | 62.4802 | 112.6284 | 139.4465 | 175.7041 |

Table 9: Reconstruction Loss ($\downarrow$) for Fashion-MNIST

| Initialization Method | 16 | | 32 | | 64 | | 128 | | 256 | | 512 | |
|---|---|---|---|---|---|---|---|---|---|---|---|---|
| | Learnable | Fixed | Learnable | Fixed | Learnable | Fixed | Learnable | Fixed | Learnable | Fixed | Learnable | Fixed |
| Gaussian ($\mu = 0, \sigma = 0.02$) | 0.0449 | 0.0475 | 0.0278 | 0.0359 | 0.0144 | 0.0205 | 0.0044 | 0.0241 | 0.0053 | 0.0587 | 0.0426 | 0.0937 |
| Gaussian ($\mu = 0, \sigma = 0.5$) | 0.0457 | 0.0466 | 0.0279 | 0.0334 | 0.0148 | 0.0172 | 0.0049 | 0.0108 | **0.0037** | 0.0186 | **0.0169** | 0.0397 |
| Gaussian ($\mu = 0, \sigma = 1$) | 0.0463 | 0.0474 | 0.0281 | 0.0335 | 0.0151 | 0.0176 | 0.0049 | 0.0114 | 0.0067 | 0.0585 | 0.0404 | 0.0776 |
| Gaussian ($\mu = 1, \sigma = 0.02$) | 0.0456 | 0.0744 | 0.0282 | 0.0950 | 0.0502 | 0.0813 | 0.1028 | 0.1435 | 0.1027 | 0.2329 | 0.1026 | 0.2791 |
| Gaussian ($\mu = 1, \sigma = 0.5$) | 0.0455 | 0.0852 | 0.0277 | 0.0885 | 0.0145 | 0.0961 | 0.0090 | 0.1570 | 0.0138 | 0.2348 | 0.0394 | 0.1347 |
| Gaussian ($\mu = 1, \sigma = 1$) | 0.0461 | 0.0755 | 0.0280 | 0.0814 | 0.0145 | 0.0585 | 0.0046 | 0.1819 | 0.0053 | 0.2909 | 0.0444 | 0.2426 |
| Gaussian ($\mu = -1, \sigma = 0.02$) | 0.1031 | 0.1031 | 0.1030 | 0.1029 | 0.1028 | 0.1027 | 0.1026 | 0.1026 | 0.1028 | 0.1027 | 0.1027 | 0.1026 |
| Gaussian ($\mu = -1, \sigma = 0.5$) | 0.0462 | 0.1031 | 0.0301 | 0.1029 | **0.0134** | 0.0957 | 0.0330 | 0.0979 | 0.0755 | 0.0992 | 0.0750 | 0.0992 |
| Gaussian ($\mu = -1, \sigma = 1$) | 0.0454 | 0.0810 | 0.0287 | 0.0764 | 0.0152 | 0.0769 | 0.0046 | 0.0862 | 0.0050 | 0.0913 | 0.0386 | 0.0915 |
| Orthogonal (Gain = 0.5) | 0.0456 | 0.0462 | 0.0281 | 0.0333 | 0.0145 | 0.0184 | 0.0044 | 0.0144 | 0.0052 | 0.0444 | 0.0438 | 0.0870 |
| Orthogonal (Gain = 1) | 0.0461 | 0.0459 | 0.0283 | 0.0322 | 0.0147 | 0.0171 | 0.0046 | 0.0102 | 0.0047 | 0.0162 | 0.0457 | 0.0370 |
| Orthogonal (Gain = 1.5) | 0.0461 | 0.0459 | 0.0283 | 0.0322 | 0.0147 | 0.0171 | 0.0046 | 0.0102 | 0.0047 | 0.0162 | 0.0457 | 0.0370 |
| Uniform (-0.02, 0.02) | 0.0455 | 0.0489 | 0.0278 | 0.0372 | 0.0144 | 0.0234 | **0.0043** | 0.0553 | 0.0052 | 0.1027 | 0.0444 | 0.1026 |
| Uniform (-0.5, 0.5) | 0.0452 | 0.0468 | 0.0279 | 0.0330 | 0.0149 | 0.0172 | 0.0048 | 0.0105 | 0.0040 | 0.0162 | 0.0220 | 0.0343 |
| Uniform (-1 to 1) | 0.0457 | 0.0472 | 0.0276 | 0.0334 | 0.0151 | 0.0171 | 0.0049 | 0.0107 | 0.0039 | 0.0182 | 0.0264 | 0.0475 |
| Uniform (-1.5, 1.5) | 0.0453 | 0.0473 | 0.0284 | 0.0329 | 0.0149 | 0.0170 | 0.0049 | 0.0109 | 0.0060 | 0.0207 | 0.0401 | 0.0899 |
| Uniform (0, 1) | 0.0457 | 0.0450 | **0.0275** | 0.0290 | 0.0145 | 0.0412 | 0.0099 | 0.0847 | 0.0520 | 0.2173 | 0.0779 | 0.2294 |
| Xavier Normal (Gain = 0.5) | 0.0452 | 0.0473 | 0.0282 | 0.0349 | 0.0146 | 0.0196 | 0.0044 | 0.0285 | 0.0055 | 0.0789 | 0.0412 | 0.1026 |
| Xavier Normal (Gain = 1) | 0.0452 | 0.0466 | 0.0278 | 0.0338 | 0.0143 | 0.0180 | 0.0044 | 0.0119 | 0.0053 | 0.0261 | 0.0427 | 0.0602 |
| Xavier Normal (Gain = 1.5) | **0.0448** | 0.0466 | 0.0279 | 0.0335 | 0.0146 | 0.0174 | 0.0044 | 0.0112 | 0.0048 | 0.0186 | 0.0447 | 0.0439 |
| Xavier Uniform (Gain = 0.5) | 0.0453 | 0.0480 | 0.0285 | 0.0367 | 0.0145 | 0.0288 | 0.0045 | 0.0795 | 0.0058 | 0.1027 | 0.0404 | 0.1026 |
| Xavier Uniform (Gain = 1) | 0.0461 | 0.0465 | 0.0286 | 0.0339 | 0.0145 | 0.0197 | 0.0044 | 0.0196 | 0.0058 | 0.0527 | 0.0398 | 0.0858 |
| Xavier Uniform (Gain = 1.5) | 0.0459 | 0.0469 | 0.0280 | 0.0334 | 0.0143 | 0.0181 | 0.0045 | 0.0107 | 0.0054 | 0.0239 | 0.0455 | 0.0518 |
| Kaiming Normal (Mode = fan_in) | 0.0458 | 0.0498 | 0.0280 | 0.0370 | 0.0149 | 0.0172 | 0.0044 | 0.0190 | 0.0049 | 0.0248 | 0.0490 | 0.0381 |
| Kaiming Normal (Mode = fan_out) | 0.0458 | 0.0463 | 0.0300 | 0.0337 | 0.0144 | 0.0319 | 0.0044 | 0.0494 | 0.0063 | 0.0482 | 0.0416 | 0.0631 |
| Kaiming Uniform (Mode = fan_in) | 0.0454 | 0.0471 | 0.0280 | 0.0331 | 0.0149 | 0.0173 | 0.0046 | 0.0102 | 0.0052 | 0.0162 | 0.0462 | 0.0366 |
| Kaiming Uniform (Mode = fan_out) | 0.0453 | 0.0466 | 0.0284 | 0.0337 | 0.0144 | 0.0193 | 0.0044 | 0.0150 | 0.0058 | 0.0403 | 0.0423 | 0.0680 |
| Hadamard | 0.0460 | 0.0462 | 0.0281 | 0.0327 | 0.0149 | 0.0170 | 0.0046 | 0.0111 | 0.0047 | 0.0189 | 0.0449 | 0.0429 |

Table 10: SSIM (↑) for Fashion-MNIST

| Initialization Method | 16 | | 32 | | 64 | | 128 | | 256 | | 512 | |
|---|---|---|---|---|---|---|---|---|---|---|---|---|
| | Learnable | Fixed | Learnable | Fixed | Learnable | Fixed | Learnable | Fixed | Learnable | Fixed | Learnable | Fixed |
| Gaussian ($\mu = 0, \sigma = 0.02$) | **0.4464** | 0.2951 | 0.6329 | 0.4145 | 0.8036 | 0.6093 | 0.9345 | 0.5732 | 0.9349 | 0.2725 | 0.4938 | 0.0664 |
| Gaussian ($\mu = 0, \sigma = 0.5$) | 0.4425 | 0.3733 | 0.6280 | 0.5175 | 0.7904 | 0.7467 | 0.9241 | 0.8434 | 0.9428 | 0.7560 | **0.7848** | 0.5023 |
| Gaussian ($\mu = 0, \sigma = 1$) | 0.4338 | 0.3715 | 0.6262 | 0.5295 | 0.7838 | 0.7593 | 0.9237 | 0.8483 | 0.9167 | 0.4837 | 0.5080 | 0.2949 |
| Gaussian ($\mu = 1, \sigma = 0.02$) | 0.4455 | 0.3213 | 0.6286 | 0.2799 | 0.4947 | 0.3182 | 0.0548 | 0.0792 | 0.0548 | 0.0230 | 0.0549 | 0.0033 |
| Gaussian ($\mu = 1, \sigma = 0.5$) | 0.4417 | 0.2980 | 0.6317 | 0.3367 | 0.7997 | 0.2298 | 0.8820 | 0.0583 | 0.8275 | 0.0115 | 0.5195 | 0.0658 |
| Gaussian ($\mu = 1, \sigma = 1$) | 0.4377 | 0.3136 | 0.6332 | 0.3444 | 0.7930 | 0.4885 | 0.9326 | 0.0343 | 0.9344 | 0.0196 | 0.4727 | 0.0075 |
| Gaussian ($\mu = -1, \sigma = 0.02$) | 0.0549 | 0.0548 | 0.0547 | 0.0546 | 0.0546 | 0.0547 | 0.0548 | 0.0547 | 0.0548 | 0.0549 | 0.0548 | 0.0548 |
| Gaussian ($\mu = -1, \sigma = 0.5$) | 0.4045 | 0.0547 | 0.5907 | 0.0545 | **0.8176** | 0.0773 | 0.6446 | 0.0674 | 0.2575 | 0.0586 | 0.1791 | 0.0584 |
| Gaussian ($\mu = -1, \sigma = 1$) | 0.4358 | 0.1423 | 0.6167 | 0.1742 | 0.7819 | 0.1384 | 0.9289 | 0.1041 | 0.9322 | 0.0663 | 0.5379 | 0.0662 |
| Orthogonal (Gain = 0.5) | 0.4440 | 0.3395 | 0.6288 | 0.4761 | 0.8004 | 0.6646 | 0.9337 | 0.7111 | 0.9348 | 0.3707 | 0.4799 | 0.0744 |
| Orthogonal (Gain = 1) | 0.4357 | 0.3787 | 0.6260 | 0.5247 | 0.7916 | 0.7134 | 0.9327 | 0.8101 | 0.9410 | 0.7336 | 0.4598 | 0.4862 |
| Orthogonal (Gain = 1.5) | 0.4357 | 0.3787 | 0.6260 | 0.5247 | 0.7916 | 0.7134 | 0.9327 | 0.8101 | 0.9410 | 0.7336 | 0.4598 | 0.4862 |
| Uniform (-0.02, 0.02) | 0.4396 | 0.2737 | 0.6333 | 0.3895 | 0.8050 | 0.5556 | 0.9284 | 0.2758 | 0.9348 | 0.0548 | 0.4776 | 0.0548 |
| Uniform (-0.5, 0.5) | 0.4427 | 0.3753 | 0.6323 | 0.5231 | 0.7897 | 0.7343 | 0.9284 | 0.8363 | **0.9460** | 0.7717 | 0.7284 | 0.5615 |
| Uniform (-1, 1) | 0.4389 | 0.3807 | **0.6375** | 0.5327 | 0.7876 | 0.7523 | 0.9230 | 0.8509 | 0.9438 | 0.7638 | 0.6684 | 0.3981 |
| Uniform (-1.5, 1.5) | 0.4430 | 0.3821 | 0.6241 | 0.5462 | 0.7889 | 0.7622 | 0.9219 | 0.8536 | 0.9255 | 0.7303 | 0.5132 | 0.2118 |
| Uniform (0, 1) | 0.4374 | 0.3996 | 0.6351 | 0.6209 | 0.7993 | 0.6218 | 0.8746 | 0.2312 | 0.4861 | 0.0238 | 0.2292 | 0.0095 |
| Xavier Normal (Gain = 0.5) | 0.4411 | 0.3173 | 0.6296 | 0.4445 | 0.7994 | 0.6344 | 0.9350 | 0.5284 | 0.9327 | 0.1357 | 0.5104 | 0.0548 |
| Xavier Normal (Gain = 1) | 0.4456 | 0.3389 | 0.6320 | 0.4728 | 0.8023 | 0.6797 | 0.9321 | 0.7494 | 0.9360 | 0.5988 | 0.4917 | 0.2412 |
| Xavier Normal (Gain = 1.5) | 0.4458 | 0.3500 | 0.6311 | 0.4899 | 0.7949 | 0.7017 | **0.9355** | 0.7771 | 0.9402 | 0.6963 | 0.4628 | 0.4038 |
| Xavier Uniform (Gain = 0.5) | 0.4378 | 0.3066 | 0.6283 | 0.4256 | 0.8012 | 0.5158 | 0.9331 | 0.1257 | 0.9288 | 0.0548 | 0.5182 | 0.0548 |
| Xavier Uniform (Gain = 1) | 0.4253 | 0.3397 | 0.6259 | 0.4783 | 0.8021 | 0.6419 | 0.9343 | 0.6502 | 0.9291 | 0.2851 | 0.5171 | 0.0902 |
| Xavier Uniform (Gain = 1.5) | 0.4345 | 0.3533 | 0.6322 | 0.4980 | 0.8019 | 0.6806 | 0.9337 | 0.7761 | 0.9344 | 0.6301 | 0.4621 | 0.3056 |
| Kaiming Normal (Mode = fan_in) | 0.4361 | 0.3183 | 0.6272 | 0.4520 | 0.7925 | 0.7176 | 0.9332 | 0.6815 | 0.9386 | 0.6253 | 0.4349 | 0.4780 |
| Kaiming Normal (Mode = fan_out) | 0.4175 | 0.3459 | 0.5947 | 0.4783 | 0.8013 | 0.5111 | 0.9315 | 0.3530 | 0.9234 | 0.4287 | 0.5001 | 0.2521 |
| Kaiming Uniform (Mode = fan_in) | 0.4424 | 0.3689 | 0.6310 | 0.5162 | 0.7916 | 0.7187 | 0.9319 | 0.8058 | 0.9370 | 0.7365 | 0.4569 | 0.4953 |
| Kaiming Uniform (Mode = fan_out) | 0.4413 | 0.3454 | 0.6253 | 0.4804 | 0.8045 | 0.6519 | 0.9347 | 0.7113 | 0.9286 | 0.4211 | 0.4995 | 0.1642 |
| Hadamard | 0.4388 | 0.3718 | 0.6296 | 0.5134 | 0.7921 | 0.7179 | 0.9313 | 0.7922 | 0.9405 | 0.6963 | 0.4690 | 0.4126 |

Table 11: FID (↓) for Fashion-MNIST

| Initialization Method | 16 | | 32 | | 64 | | 128 | | 256 | | 512 | |
|---|---|---|---|---|---|---|---|---|---|---|---|---|
| | Learnable | Fixed | Learnable | Fixed | Learnable | Fixed | Learnable | Fixed | Learnable | Fixed | Learnable | Fixed |
| Gaussian ($\mu = 0, \sigma = 0.02$) | 178.4276 | 218.7172 | 133.8405 | 197.8785 | 79.0841 | 126.2222 | 24.4798 | 131.4325 | 18.4275 | 233.8625 | 141.1779 | 319.4675 |
| Gaussian ($\mu = 0, \sigma = 0.5$) | 171.5091 | 210.4390 | 136.5693 | 175.8949 | 80.9980 | 98.8907 | 28.7288 | 57.7473 | 18.0161 | 89.5628 | **67.2428** | 163.3966 |
| Gaussian ($\mu = 0, \sigma = 1$) | 172.3919 | 211.9167 | 135.2910 | 167.4778 | 82.8584 | 92.0028 | 29.7082 | 52.4620 | 23.3368 | 180.9148 | 153.6093 | 276.7094 |
| Gaussian ($\mu = 1, \sigma = 0.02$) | 173.9740 | 195.8672 | 138.0503 | 169.6501 | 192.9372 | 254.9356 | 352.6039 | 379.8038 | 353.5087 | 415.9422 | 355.5287 | 409.5339 |
| Gaussian ($\mu = 1, \sigma = 0.5$) | **170.2878** | 173.4771 | 138.1840 | 166.6962 | 78.1149 | 260.6100 | 41.6254 | 399.3627 | 57.2834 | 378.6912 | 146.8364 | 468.2354 |
| Gaussian ($\mu = 1, \sigma = 1$) | 170.8678 | 187.8037 | 133.6235 | 165.7766 | 80.3770 | 184.5602 | 25.3015 | 390.4048 | 19.2555 | 411.6939 | 148.5768 | 430.0127 |
| Gaussian ($\mu = -1, \sigma = 0.02$) | 350.5009 | 352.1599 | 348.1890 | 349.9350 | 348.6937 | 348.2652 | 349.4396 | 348.0315 | 352.0145 | 351.9135 | 353.1674 | 352.1848 |
| Gaussian ($\mu = -1, \sigma = 0.5$) | 198.2017 | 355.3506 | 157.1520 | 352.8964 | 83.9886 | 323.8154 | 129.1395 | 326.2354 | 266.9524 | 348.3002 | 253.6508 | 349.4343 |
| Gaussian ($\mu = -1, \sigma = 1$) | 182.2203 | 232.7002 | 143.3895 | 220.9111 | 82.6830 | 234.8540 | 29.5301 | 247.8729 | 20.8915 | 338.9296 | 137.1517 | 337.1287 |
| Orthogonal (Gain = 0.5) | 176.2405 | 213.4522 | 138.0169 | 177.5687 | 79.5886 | 119.5016 | 24.5925 | 87.5695 | 17.8503 | 189.0431 | 141.0744 | 306.9878 |
| Orthogonal (Gain = 1) | 171.1702 | 210.4954 | 132.9115 | 170.7393 | 80.9021 | 109.7682 | 25.1612 | 64.8384 | 15.9699 | 105.2417 | 150.6491 | 161.4446 |
| Orthogonal (Gain = 1.5) | 171.1702 | 210.4954 | 132.9115 | 170.7393 | 80.9021 | 109.7682 | 25.1612 | 64.8384 | 15.9699 | 105.2417 | 150.6491 | 161.4446 |
| Uniform (-0.02, 0.02) | 175.2026 | 224.6855 | 135.8981 | 199.1652 | 78.9040 | 135.4349 | 26.1417 | 231.1302 | 19.2092 | 349.7004 | 144.7068 | 353.2435 |
| Uniform (-0.5, 0.5) | 173.6673 | 207.5652 | 136.0056 | 174.0267 | 80.0325 | 104.0387 | 28.3653 | 59.9937 | 16.3568 | 84.9445 | 90.0399 | 140.6081 |
| Uniform (-1, 1) | 176.8569 | 209.5528 | **132.0886** | 174.8881 | 81.7074 | 96.6367 | 29.9206 | 55.3884 | 17.6826 | 85.3822 | 106.0517 | 195.8420 |
| Uniform (-1.5, 1.5) | 176.9801 | 208.0298 | 135.2002 | 163.6716 | 82.3362 | 90.5122 | 29.4016 | 51.9878 | 20.2563 | 98.4423 | 147.2280 | 289.9079 |
| Uniform (0, 1) | 172.3307 | 211.0640 | 134.1532 | 151.2473 | **77.9346** | 126.0953 | 42.6135 | 324.7710 | 187.1713 | 402.9615 | 279.4473 | 419.7646 |
| Xavier Normal (Gain = 0.5) | 176.7011 | 216.8205 | 138.2731 | 185.9724 | 80.6514 | 124.1608 | **24.4654** | 149.4544 | 19.3133 | 287.4142 | 140.2415 | 355.5469 |
| Xavier Normal (Gain = 1) | 171.8320 | 209.2904 | 135.0339 | 180.2099 | 78.7730 | 116.8141 | 26.5568 | 77.3584 | 17.2359 | 136.0253 | 146.5963 | 224.1291 |
| Xavier Normal (Gain = 1.5) | 173.0749 | 205.3336 | 136.4374 | 176.0217 | 82.7547 | 111.3343 | 24.6124 | 72.9501 | 15.8888 | 112.7811 | 149.3688 | 182.3617 |
| Xavier Uniform (Gain = 0.5) | 182.4055 | 224.1428 | 136.1948 | 192.6981 | 78.7905 | 163.9333 | 25.4300 | 310.9050 | 19.6205 | 354.9524 | 138.8488 | 356.9361 |
| Xavier Uniform (Gain = 1) | 186.6149 | 216.3850 | 136.1653 | 183.9397 | 79.1636 | 129.2100 | 26.8881 | 117.3258 | 19.8907 | 222.9482 | 140.5917 | 307.2277 |
| Xavier Uniform (Gain = 1.5) | 173.3115 | 213.9354 | 132.8233 | 176.4165 | 79.3204 | 119.7971 | 26.1742 | 71.9950 | 17.3586 | 130.5526 | 150.1506 | 212.0603 |
| Kaiming Normal (Mode = fan_in) | 175.7812 | 227.0953 | 140.5513 | 189.9054 | 80.4035 | 105.3271 | 25.3717 | 109.5506 | 16.4685 | 128.7356 | 151.4423 | 164.2091 |
| Kaiming Normal (Mode = fan_out) | 192.7411 | 209.5611 | 149.0085 | 176.3418 | 81.1644 | 169.0999 | 25.0696 | 203.0001 | 22.2838 | 197.8771 | 143.8145 | 238.8719 |
| Kaiming Uniform (Mode = fan_in) | 175.2201 | 206.2546 | 132.4174 | 175.1433 | 80.4289 | 108.4591 | 25.0194 | 68.2233 | 15.7122 | 100.8423 | 145.9667 | 158.9615 |
| Kaiming Uniform (Mode = fan_out) | 177.8347 | 216.5534 | 133.4456 | 182.2388 | 78.6883 | 127.3091 | 24.6866 | 97.3891 | 19.6215 | 185.9586 | 142.7112 | 247.3069 |
| Hadamard | 171.0226 | 202.5676 | 134.6836 | 170.8583 | 80.9484 | 105.2641 | 25.6911 | 68.0015 | **15.4854** | 112.6257 | 144.3757 | 176.1968 |

