# OpenReview forum: "Decoding Projections From Frozen Random Weights in Autoencoders: What Information Do They Encode?"
_NeurIPS.cc/2025/Workshop/UniReps — UniReps2025_

### Official Review · Reviewer_rxDe · 2025-09-06
**Empirical study aimed at establishing the quality of representations from autoencoders with untrained encoders**

**Confidence:** 4

**Review:**

**Summary:**
- The authors propose to study the information content in the representations of a randomly initialized convolutional autoencoder empirically by tracking perceptual metrics (SSIM and FID) when the encoder weights are fixed vs learnable across different latent dimensionalities and initialization schemes.
- The overall setup shows promise, but the execution is lacking and the results fail to fully back the authors' claims.
- While I have voted for the work to not be accepted, I am not strongly opposed to it being presented at the workshop. However, I do believe that it has the potential to be much better and thorough, and urge the authors to consider conducting more analyses and give the following suggestions and comments due consideration.

**Strengths:**
- The intent behind the work itself is very relevant, and if studied correctly, the questions motivating the work could shed some interesting insight on how the architectural aspects of the models we use themselves preserve/erode information within representations and bias them in certain ways.

**Weaknesses:**
- Limited insight. I believe this is a consequence of both, i) the style and extent of the experiments, i.e., the choice of datasets, architecture, evaluation method, and ii) novelty w.r.t. prior literature. While I appreciate the current take-aways of the work, it is worth asking if efficiency in the number of parameters/convergence rate or architecture selection in terms of latent space dimensionality for an autoencoder is the intended payoff when it was motivated with the broad question, "How useful are the weights of an untrained NN and what information do they capture?"
- The results of Fig. 1 are somewhat cherry-picked given that these runs were selected from a bunch of different ones and that we don't consistently find an $R^2$ on the order of 0.8.
    - To that end, I would've appreciated seeing a statistically significant estimate on the correlation between the learnable and fixed architectures' metrics for the different initialization schemes, as it would've given us some insight into which intialization schema best support the notion of linear tracking between the fixed and learnable models' metrics.
    - I think it would also be interesting to study the setup in an additional "Tied" case where the weights of the decoder are constrained to be the (conv-)transpose of the encoder (i.e., tied), thereby forcing the model to reconstruct only from what the encoder makes linearly recoverable. This would isolate how "invertible/information-preserving" the encoder's representations are, without giving the decoder extra degrees of freedom to compensate. Comparing how far the tied model must move the encoder from its initial weights to get good reconstructions would tell us how much reconstructive information was already present in the original encoder. I anticipate the results from such an experiment again shedding some light on which initializations are better suited distributionally to retaining more or less information in the representations.
- I am also not sure how much stock to put in a measure like SSIM and FID for the CIFAR-10 dataset (with things being ever so slightly better for the other datasets). My feeling is that these measures are likely very susceptible to pretty small changes and misregistrations given the scale they're at.
    - I also think the current set of measures would be complemented by measuring the effective JL-style perceptual distortment of the latent representations, i.e., seeing which initialization distributions, architectures, etc. minimize $\mathbb{E}[\delta]$ where
$$\delta_{ij} = \frac{||\phi(x_i) - \phi(x_j)|| - \lambda d(\hat{x}_i - \hat{x}_j)}{d(\hat{x}_i - \hat{x}_j)}$$
for all $x_i,x_j \in \mathcal{D}$ and $d$ refers to the distance in terms of some perceptual metric. This again, I believe could help separate representation information from decoder capacity.
    - Training a linear probe on the latents to predict per-image quality (FID or something a little less noisy that SSIM) could also be informative. Strong test-time predictability—ideally transferring across latent sizes/inits—would show that the encoder linearly encodes reconstructability, separating genuine representation geometry from decoder/optimization artifacts; weak predictability would indicate the opposite.
- W.r.t. Fig. 2 and the claim that "random" representations can serve as good baselines, I will note that while they generally follow the same trends, the fixed representations start to degrade sooner than the learnt representations achieve their "peak" which I think makes things a little misleading.

**Misc. Comments:**
- In the first line of the abstract, "neural networks without gradient updates" is confusing -- It might be better to state that the authors wish to study the inherent architectural/inductive biases of the networks. The way it is currently phrased one might confuse their intent with wanting to study neural networks which have been optimized with derivative-free procedures, e.g., hill climbing methods or genetic algorithms.
- Similar comment for the first line(s) of the introduction as well.
- Line 9: "... learn task-specific ~input~ representations by defining a loss..."
- Deep Image Prior and similar works should've been cited in the main text as well, not just the appendix. Expanding on how these studies are different from the one conducted by the authors would be helpful in strengthening the autors' contributions as well (I would recommend doing this instead of providing some of the logistical/methodological details in lines 70-80 and moving those to the appendix instead).

**Score:**

2

**Topic Fit:**

2

---

### Official Review · Reviewer_pjA8 · 2025-09-09

**Confidence:** 4

**Review:**

This paper contains an empirical analysis to compare the representational quality of two types of autoencoder, one where the model is trained end-to-end, and another, where the encoder is frozen with randomly initialized weights and only the decoder is trained. While the full autoencoder generally outperforms the random encoder model, the authors find that the random encoder model can serve as a useful baseline to evaluate hyperparameters, as it is more efficient to train and follows similar trends as the learnable model.

 ## Strengths
- The paper empirically investigates representations of random neural networks. To my knowledge, random neural networks are mostly explored from a theoretical perspective with only limited empirical results.
- The authors clearly describe their setting, and the experimental setup is easy to follow.
## Weaknesses
- The biggest weakness of this work is the scope of the experiments. While the authors investigated many combinations of hyperparameters for the weight initializations, it would have been much more interesting to see the impact of random weights on different architectures. How do results compare between MLPs, CNNs, Transformers, or RNNs? Can we find any specific patterns there? The results are also conducted on relatively small-scale datasets with CIFAR-10/100 and Fashion-MNIST. What happens if you repeat the experiments on larger or higher resolution datasets?
- Larger latent spaces lead to less of an information bottleneck, typically leading to better reconstructions. Here, it is unclear why reconstruction errors increase for large latent dimensionalities. Could there be an issue with overfitting?
- The authors claim that random networks can be a useful tool for efficiently performing architecture selection on a new dataset. It would be interesting to include an experiment where you show that the best architecture for a random network matches the best architecture when training the full autoencoder to back up this claim.

**Score:**

3

**Topic Fit:**

2

---

### Official Review · Reviewer_ti8v · 2025-09-13
**AutoEncoders with random encoder might be as capable as its trained counterpart**

**Confidence:** 3

**Review:**

**Summary**
The work investigates the capability of AutoEncoders (AEs) with the weight of the encoder random and how factors (e.g., datasets, width, and intialization scheme) influences the capability. Preliminary results on small vision datasets (C10, C100, FMNIST) demonstrates that,  if parameters are chosed properly, AEs with random weights could achieve the same performance as their learnable counterparts. The insight could be the basis for designing more efficient architectures.


**Comments**

- The manuscript and their methodology (as well as experimental details in the appendix) are clearly written.
- I appreciate the detailed related work, though I can't judge whether it is complete as I am not in the field.

(minor)
- imho, fig 1 is quite difficult to interpret. It might be easier if the plot uses the same scale for the x- and y- axes. Then, the point on the diagonal line would be of interest
- perhaps, the authors might consider add err bars to fig 1 and 2

**Questions**
- I would be interested in seeing similar experiments on NNs for supervision.
- In the context of mechanistic interpretability,  sparse AEs are one of the main tools to discover meaningful structures (or disentangling concepts) in LLMs. I would be curious to see what happens if we use random weight for the encoder. Will we find any structure? If the answer is positive, it could have implication on how we interpret the results from sparse AEs.

**Score:**

4

**Topic Fit:**

3

---

### Official Review · Reviewer_idY2 · 2025-09-16
**Autoencoders With Fixed Random Weights Provide Useful Baselines Yet Remain Limited in Practice**

**Confidence:** 3

**Review:**

This paper investigates what information is preserved when using randomly initialized, frozen encoder weights in autoencoders. The paper compares two settings in autoencoders: one with learnable encoders and one with frozen random encoders (only the decoder is trained). Experiments across multiple datasets, latent dimensions, and initialization schemes show that even fixed random encoders can preserve broad structural information. The authors argue that frozen random encoders can serve as useful baselines for representation learning.

Strengths:
1. Novel empirical contribution: While prior work focused mainly on theoretical analysis of random weights (e.g., Johnson-Lindenstrauss lemma), this is one of the first systematic empirical studies that directly measures perceptual and quantitative outcomes of frozen random autoencoders.
2. Positioning as a baseline: The proposal that random encoders can serve as a baseline is valuable for reducing computational overhead and for interpretability in neural network research.

Weaknesses:
1. Questionable value as a baseline. While the authors argue that frozen random encoders should be considered a baseline, the practical usefulness of this is limited. Existing baselines such as PCA, random features, already capture the idea of architecture-driven structure without training. The proposed baseline does not clearly offer additional interpretive or benchmarking value, since its reconstructions remain coarse and it provides little insight for downstream tasks beyond reconstruction quality.
2. Overemphasis on reconstruction quality: The evaluation is almost entirely based on how well inputs are reconstructed. But good reconstructions don’t necessarily imply useful representations, they could just reflect pixel-level correlations rather than meaningful features.
3. Insufficient exploration of initialization schemes: While many initializations are tested, the paper doesn’t fully analyze why some perform differently. This makes the findings more catalog-like than explanatory.
4. Lack of downstream evaluation: The study focuses on reconstruction tasks, but does not test whether representations from frozen random encoders are useful for classification, clustering, or other downstream applications where meaningful structure matters.
5. Limited scope of datasets: Only three datasets are tested. While useful, they may not capture the full diversity of real-world settings. Larger-scale datasets (e.g., ImageNet) would strengthen the claim of general applicability.

**Score:**

2

**Topic Fit:**

3